# Forming intracluster gas in a galaxy protocluster at a redshift of 2.16

Luca Di Mascolo[1,2,3 ✉], Alexandro Saro[1,2,3,4], Tony Mroczkowski[5], Stefano Borgani[1,2,3,4], Eugene Churazov[6,7], Elena Rasia[2,3], Paolo Tozzi[8], Helmut Dannerbauer[9,10], Kaustuv Basu[11], Christopher L. Carilli[12], Michele Ginolfi[5,13], George Miley[14], Mario Nonino[2], Maurilio Pannella[1,2,3], Laura Pentericci[15] & Francesca Rizzo[16,17]

Galaxy clusters are the most massive gravitationally bound structures in the Universe, comprising thousands of galaxies and pervaded by a diffuse, hot intracluster medium (ICM) that dominates the baryonic content of these systems. The formation and evolution of the ICM across cosmic time[1] is thought to be driven by the continuous accretion of matter from the large-scale filamentary surroundings and energetic merger events with other clusters or groups. Until now, however, direct observations of the intracluster gas have been limited only to mature clusters in the later three-quarters of the history of the Universe, and we have been lacking a direct view of the hot, thermalized cluster atmosphere at the epoch when the first massive clusters formed. Here we report the detection (about $6\sigma$) of the thermal Sunyaev–Zeldovich (SZ) effect[2] in the direction of a protocluster. In fact, the SZ signal reveals the ICM thermal energy in a way that is insensitive to cosmological dimming, making it ideal for tracing the thermal history of cosmic structures[3]. This result indicates the presence of a nascent ICM within the Spiderweb protocluster at redshift $z = 2.156$, around 10 billion years ago. The amplitude and morphology of the detected signal show that the SZ effect from the protocluster is lower than expected from dynamical considerations and comparable with that of lower-redshift group-scale systems, consistent with expectations for a dynamically active progenitor of a local galaxy cluster.

To measure the SZ effect of the protocluster complex surrounding PKS 1138−262 ($z = 2.156$; commonly known as the Spiderweb galaxy), we used the Atacama Large Millimeter/submillimeter Array and obtained deep Band 3 (94.5–110.5 GHz) observations, exploiting both the main 12-m array (ALMA) and the 7-m Atacama Compact Array (ACA).

The transition from sparse protocluster complexes to mature, nearly virialized systems is a tumultuous process[1,4]. Energetic events—for example, infall and accretion of the diffuse medium from surrounding filaments, mergers with substructures and feedback from active galactic nuclei (AGN)—affect the regularity of the assembling ICM, with simulations predicting that their effects could persist for more than a Hubble time[5]. From an observational point of view, this implies that building a simple, analytical model describing the morphology of the disturbed proto-ICM is not a trivial task. Further, the possibility of mapping the ICM within galaxy protoclusters relies on our ability to separate the SZ footprint from any contaminating sources within the field. In fact, the Spiderweb galaxy harbours a powerful AGN[6,7], with associated hybrid-morphology[8,9] jets extending over scales of about 100 kpc. The result is that the millimetre-wavelength continuum signal in the direction of the Spiderweb protocluster is dominated by the emission from the central radio galaxy. In the available Band 3 ACA data (probing the SZ signal thanks to its ability to recover larger angular scales), we measure a peak surface brightness for the continuum emission from the central radio source of $2.39 \pm 0.12$ mJy beam$^{-1}$. By assuming that, also at the Spiderweb redshift, the total mass and volume-integrated SZ signal scale covariantly (as established theoretically and empirically for low-$z$ clusters; see ref. [10] for a review), we find that such a flux estimate is at least an order of magnitude larger than the absolute amplitude of the peak SZ signal expected for the Spiderweb protocluster ($\lesssim 0.2$ mJy beam$^{-1}$) computed by considering an upper limit of about $10^{14}\,M_\odot$ for the mass of the system[11-15] (with $M_\odot$ denoting the mass of the Sun). This large difference in the dynamic range of the surface brightness of the extended structure of the Spiderweb galaxy and of the underlying SZ effect limits the possibility of performing a robust separation of the two signals through standard imaging techniques. Thus, to handle the above complexities and obtain a statistically robust detection of the SZ signal, we need to rely on simplifying assumptions. We thus assume that the SZ signal is generated by a spherically symmetric ICM distribution and that the extended radio source can be described by a collection of point-like components

[1]Astronomy Unit, Department of Physics, University of Trieste, Trieste, Italy. [2]INAF - Osservatorio Astronomico di Trieste, Trieste, Italy. [3]IFPU - Institute for Fundamental Physics of the Universe, Trieste, Italy. [4]INFN - Sezione di Trieste, Trieste, Italy. [5]European Southern Observatory (ESO), Garching, Germany. [6]Max-Planck-Institut für Astrophysik (MPA), Garching, Germany. [7]Space Research Institute, Moscow, Russia. [8]INAF - Osservatorio Astrofisico di Arcetri, Firenze, Italy. [9]Instituto de Astrofísica de Canarias (IAC), La Laguna, Tenerife, Spain. [10]Departamento de Astrofísica, Universidad de La Laguna, La Laguna, Tenerife, Spain. [11]Argelander Institute for Astronomy, University of Bonn, Bonn, Germany. [12]National Radio Astronomy Observatory, Socorro, NM, USA. [13]Dipartimento di Fisica e Astronomia, Università di Firenze, Sesto Fiorentino, Italy. [14]Leiden Observatory, Leiden University, Leiden, The Netherlands. [15]INAF - Osservatorio Astronomico di Roma, Monte Porzio Catone, Italy. [16]Cosmic Dawn Center (DAWN), Copenhagen, Denmark. [17]Niels Bohr Institute, University of Copenhagen, Copenhagen, Denmark. ✉e-mail: luca.dimascolo@units.it

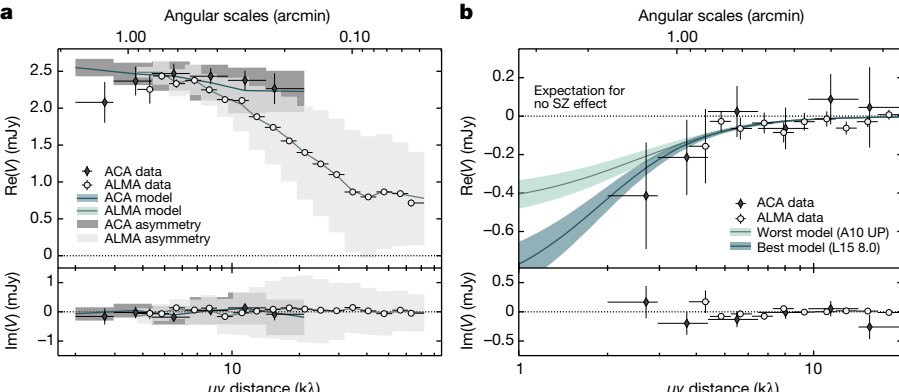

**Fig. 1 | Binned *uv* profiles of the Band 3 ALMA and ACA data. a**, Comparison of the real (Re) and imaginary (Im) parts of the visibilities (*V*) for the binned ALMA and ACA data, and for the radio-source model. The solid lines represent the corresponding median *uv* profile for each visibility set obtained by marginalizing over different numbers of point-like components (see Methods). The model uncertainties are computed accordingly but, despite being plotted, are too small to be visible. The shaded regions denote the standard deviation of the azimuthal variation of the model amplitude in each *uv* bin, owing to the elongated morphology of the radio galaxy and the asymmetric visibility patterns. The systematic drop in the real component of the visibilities at short *uv* distances (that is, large angular scales) provides evidence for the presence of a SZ decrement towards the Spiderweb complex. We note that, as we do not see a similar deviation in the imaginary component, this cannot be ascribed merely to phase variations owing to off-centre sources. **b**, Comparison of SZ models

with the ALMA and ACA data. Shown are the worst (A10 UP; ref. [28]) and best (L15 8.0; ref. [29]) SZ profiles, intended as the models that are statistically favoured the least and the most, respectively, based on their Bayesian evidence. Before binning the data, we subtracted the median radio model from the visibilities and shifted the phase centre on the SZ centroid. The uncertainties associated with the radio model and with the SZ coordinates are propagated into the ones for the ALMA and ACA data points. The shaded regions correspond to the 68% credible intervals for each model. Also in this case, the uncertainties and median SZ profiles are marginalized over models with varying numbers of point-like components. The divergence of the two models and the increase of their uncertainties at small *uv* distances (that is, large scales) is symptomatic of the limited capabilities of ALMA+ACA in constraining fluxes above about 1.70′. For both panels, the error bars denote 1σ uncertainties.

(Methods). We then analysed the available ALMA data using a Bayesian forward modelling approach[16]. The inclusion of a model component for the SZ signal is favoured by the Bayesian evidence (that is, the normalization factor in Bayes' theorem, given by the likelihood marginalized over the prior volume and key element for performing Bayesian model selection) over the case comprising only the jet and AGN emission (Fig. 1) at an effective significance $\sigma_{eff} = 5.97 \pm 0.08$. This is estimated from the difference $\Delta \log Z$ of the logarithm of the evidences (hereafter, log-evidence) for the models with and without a SZ component and by assuming the posterior distribution to be described by a multivariate normal distribution (that is, $\sigma_{eff} = \text{sgn}(\Delta \log Z) \cdot \sqrt{2|\Delta \log Z|}$). We further note that the reported value represents a conservative lower limit on the actual significance of the SZ signal, as different prescriptions for the pressure distribution of the ICM further improve the Bayesian evidence (see Extended Data Table 2 in Methods). Given the above self-similar assumption for the pressure distribution, we estimate that the detected SZ signal would correspond to a halo with $M_{500} = (3.46^{+0.38}_{-0.43}) \times 10^{13} M_{\odot}$ and $r_{500} = 228.9^{+8.4}_{-9.5}$ kpc, in which $M_{500}$ is the total mass contained in the spherical volume of radius $r_{500}$ within which the average density is 500 times larger than the critical density of the Universe at the source redshift (here, as in the rest of the manuscript, the best-fit value and the uncertainties correspond to the 50th, and 16th and 84th percentiles of the posterior distribution function for a given model parameter, respectively). At face value, this mass constraint is much lower than the naive (and largely uncertain) expectations based on velocity-dispersion measurements previously reported in the literature[11,14,15]. We find that considering different assumptions for the ICM pressure distributions slightly relieves this tension but produce mass estimates still smaller than expected from dynamical considerations. However, comparing our estimate of the volume-integrated SZ signal with the corresponding value predicted from dynamical mass estimates provides an empirical demonstration that the detected ICM halo is probably part of an extended complex of several interacting substructures. Similar hints are observed when repeating our analysis by including several SZ components to the overall

model, without however providing conclusive and statistically meaningful results. The observed discrepancy between the SZ mass estimate and the value from velocity-dispersion measurements, in combination with the evidence for a positional offset between the Spiderweb galaxy and the SZ centroid is thus consistent with a merging scenario, in which the complex structure of the pressure distribution of the proto-ICM is not necessarily well captured by the simple analytical models used in our analysis. A thorough discussion of the potential explanations of such discrepancy, together with the assessment of potential systematic issues, can be found in Methods.

To circumvent the limitations of these analytic results and corroborate our measurements with a more realistic and physically complex model, we perform a comparison of the SZ signal that we reconstructed from the ALMA+ACA data with mock SZ observations based on cosmological hydrodynamical simulations[17,18] of galaxy protoclusters (Methods). We emphasize that the scope of this comparison is not aimed at identifying an exact simulation counterpart to the Spiderweb protocluster but rather at guiding our interpretation of the measured SZ effect by providing quantitative predictions on the overall SZ signal expected to be measurable in the observations. We generate mock ALMA+ACA observations for a set of 27 simulated massive halos ($M_{500} \gtrsim 1.3 \times 10^{13} M_{\odot}$ at $z = 2.16$) that represent progenitors of galaxy clusters with masses at redshift $z = 0$ in the range $M_{500} = (5.6–8.8) \times 10^{14} M_{\odot}$ h$^{-1}$. In Fig. 2, the resulting *uv* profiles for all the simulated clusters are compared with the radio-source-subtracted ALMA+ACA data (as shown in Fig. 1). These show an agreement between the observed SZ signal, the prediction for halos of mass $M_{500} \simeq (2–5) \times 10^{13} M_{\odot}$ and, in turn, the independent estimates of the protocluster mass from the parametric modelling. Such a result provides a straightforward assessment of the reliability of the above results and, therefore, of the mismatch of the amplitude of the observed SZ signal with the expectations from dynamical considerations.

The analysis described above required a reduction of the computational complexity by limiting the range of *uv* scales used for the model reconstruction. To obtain a high-resolution view of the Spiderweb

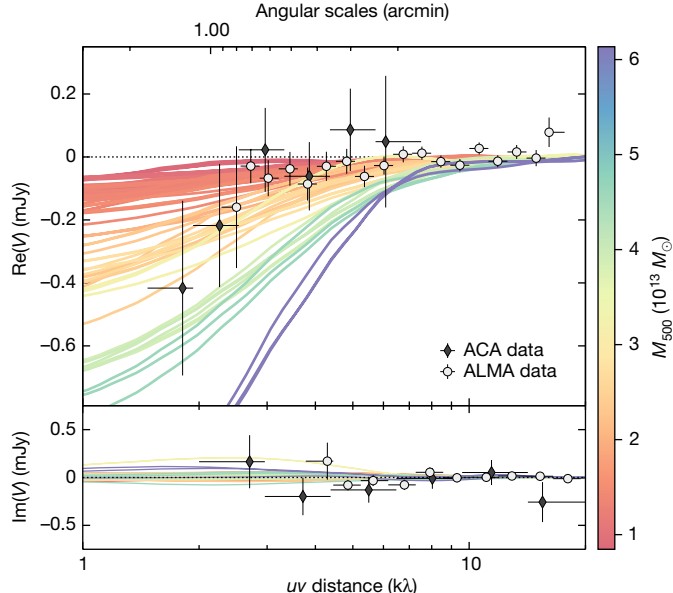

**Fig. 2 | *uv* profiles from mock SZ observations of simulated galaxy protoclusters.** Each curve represents the *uv* profile obtained for a halo extracted from the Dianoga set after projecting the corresponding SZ map onto the visibility plane, colour-coded according to their mass. For a comparison, we further plot the binned ACA and ALMA measurements from Extended Data Fig. 6. Consistent with our finding from the parametric modelling, the observed SZ signal suggests that the Spiderweb protocluster is characterized by a mass $M_{500} \simeq (2-5) \times 10^{13}\,M_\odot$. We note that this result is only weakly dependent on the cosmological model adopted in the reference simulation. As for Fig. 1, the error bars correspond to the $1\sigma$ uncertainties on the binned ALMA and ACA data. $\mathrm{Re}(V)$ and $\mathrm{Im}(V)$ are the real and imaginary parts of the visibilities ($V$), respectively, for the binned ALMA and ACA data.

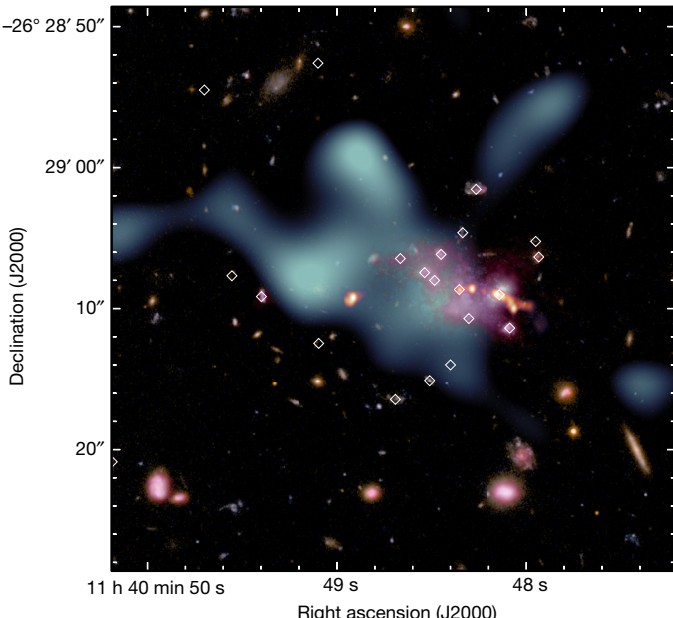

**Fig. 3 | Multiwavelength view of the Spiderweb complex.** Composite Hubble Space Telescope image based on ACS/WFC F475W and F814W data of the Spiderweb field. Overlaid are the emission from the Spiderweb galaxy and associated extended radio jet as measured by the VLA in X-band[8,9] (8–10 GHz; red), the image of the extended Lyα nebula[19,30] observed with the FORS1 instrument on the VLT (pink) and the SZ signal from a combined ALMA+ACA image (light blue). When imaging the ALMA+ACA data, we applied to the visibility weights a *uv* taper with $\sigma_{\mathrm{taper}} = 20\,\mathrm{k}\lambda$ to both suppress any noise structures on small scales and to emphasize the bulk distribution of the SZ signal from the ICM of the Spiderweb protocluster. The SZ effect offset with respect to the Spiderweb galaxy suggests that the protocluster core is undergoing a dynamically active phase. The white diamonds denote all the spectroscopically confirmed member galaxies summarized in ref. [25] and references therein.

complex and fully exploit the entirety of the dynamic range of physical scales investigated by ALMA+ACA, we apply an independent, sparse modelling approach to image the available data (see Methods for details). The resulting high-resolution image of the SZ signal obtained after subtracting the best-fit radio source from the ALMA+ACA data is provided in Fig. 3, along with a multiwavelength perspective on the Spiderweb complex (see also Extended Data Fig. 6 in Methods). Overall, the inclusion of this information in the context of the wealth of multiwavelength data associated with the Spiderweb system strongly supports an extremely dynamically active phase of protocluster formation (Methods). The multiphase environment is in fact experiencing complex interactions with the extended radio galaxy and mass accretion from the large-scale cosmic web and energetic merging events[15,19], consistent with the considerations from our parametric analysis.

Overall, the reported identification in the ALMA+ACA observations of the SZ signal from the central region of the Spiderweb complex is providing the direct indication that the system is more than a loose association of galaxies and has already started assembling its own halo of diffuse intracluster gas. Most importantly, this is providing a statistically meaningful confirmation of long-standing predictions from cosmological simulations[13] for the existence of an extended halo of thermalizing ICM within the Spiderweb protocluster, as well as of observational works, so far limited just to indirect[8,20,21] evidence or tentative detections[7,22,23]. At the same time, this detection shows that current SZ facilities could be used to effectively open a new observational window on protocluster environments. Many such systems, including more massive protoclusters, are in fact expected[4] to exist out to $z \simeq 3$. To this end, the SZ effect provides the means for obtaining a straightforward,

unambiguous confirmation of the probable progenitors of local galaxy groups and clusters, as well as of the continuing thermalization of the forming ICM in these systems or any of their parts[24]. In fact, even in the case of the extensively studied Spiderweb protocluster—for which we have spectroscopic confirmation for 112 member galaxies across the entire protocluster structure[25]—a robust identification of distinct subhalos through a spectroscopic characterization has so far been ineffective. Similarly, obtaining a robust identification of a thermal ICM component with current X-ray facilities would require a prohibitive amount of observing time, as well as an accurate separation of thermal, inverse Compton and AGN contributions to the overall X-ray signal[22,23,26,27]. We refer, for instance, to the recent Chandra study[23] of the Spiderweb protocluster (see also Extended Data Fig. 6 in Methods), which, despite being based on observations with an exposure time more than an order of magnitude larger than the ALMA+ACA measurements used in this work, provided tentative but inconclusive support to the thermal origin of a diffuse component in the X-ray emission. Identifying and performing detailed characterization of many more of the first-forming clusters during the crucial phase of vigorous relaxation and thermalization will be essential for gaining a comprehensive view of the emergence of galaxy clusters from the large-scale structure of the Universe, as well as the environmental processing of galaxies in the earliest phases of their evolution. Similarly, it will help shed light on the role of feedback mechanisms in determining the physical properties of the diffuse baryons in the proto-ICM at the very epochs when their activity is expected to peak and to have a key impact on galaxy and protocluster formation.

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

## Methods

### Cosmology

In this work, we consider a spatially flat Λ Cold Dark Matter cosmological model, with $\Omega_M = 0.30$, $\Omega_\Lambda = 0.70$ and $H_0 = 70.0$ km s$^{-1}$ Mpc$^{-1}$. At the redshift of the Spiderweb complex ($z = 2.156$), 1″ corresponds to 8.29 kpc.

### ALMA observations and reduction

An extensive observational campaign was performed during Cycle 6 to obtain a detailed view of the Spiderweb complex in Band 3 (project code: 2018.1.01526.S, principal investigator A. Saro). The data comprise measurements from the main 12-m ALMA[31] array in three different configurations (C43-1, C43-3 and C43-6), aimed at providing a high-dynamic-range view of the structure, as well as from the 7-m ACA[32] (also known as the Morita Array), complementing the ALMA observations over SZ-relevant scales (that is, over a $uv$ range of 2.2–17.5 kλ, corresponding to scales 767–97 kpc at the redshift of the Spiderweb galaxy). The spectral setup for all the configurations was tuned to cover the frequency range 94.5–110.5 GHz, split over four 2-GHz-wide spectral bands centred at 95.5 GHz, 97.5 GHz, 107.5 GHz and 109.5 GHz, respectively. In particular, the last window targets the line emission resulting from the $J = 3$–$2$ transition from the carbon monoxide (CO; rest frequency 345.796 GHz). As we are interested in modelling only the continuum component of the observed signal, we conservatively exclude all the visibilities from the spectral window expected to contain the redshifted CO $J = 3$–$2$ line. In fact, excluding only the channels corresponding to the specific emission line would provide a slight improvement in the overall statistics of the analysed data. Previous studies[33–35] of the molecular content of the nuclear region around the Spiderweb galaxy and of protocluster galaxies have shown, however, that the cold molecular gas within the Spiderweb complex is characterized by large velocity dispersion as well as broad differences in the systemic velocities of the member galaxies. Faint tails and offset components may hence contaminate the continuum signal in any channels in the proximity of the emission line, potentially affecting the model reconstruction.

Data calibration was performed in the Common Astronomy Software Applications[36] (CASA; https://casa.nrao.edu/) package version 5.4.0 using the standard reduction pipeline provided as part of the data delivery. The direct inspection of output visibility tables highlighted no clear issues with the outcome of the pipeline calibration and we therefore did not perform any extra flagging or postprocessing tuning. The resulting root-mean-square (RMS) noise levels of the observations are estimated to be 4.10 μJy beam$^{-1}$, 29.1 μJy beam$^{-1}$ and 14.2 μJy beam$^{-1}$ for the C43-1, C43-3 and C43-6 ALMA measurements, respectively, and 32.0 μJy beam$^{-1}$ for the ACA data.

To obtain better knowledge of the spectral properties of the measured signals, we further include in our analyses archival Band 4 ALMA (project code: 2015.1.00851.S, principal investigator B. Emonts) and ACA (project code 2016.2.00048.S, principal investigator B. Emonts) observations. In both cases, we use the calibrated measurement sets provided by the European ALMA Regional Centre network[37] through the calMS service[38]. The achieved RMS noise levels amount to 99.0 μJy beam$^{-1}$ and 6.29 μJy beam$^{-1}$ for the ACA and ALMA data, respectively. As with the Band 3 measurements, we exclude from our analyses the spectral windows in Band 4 covering the CO $J = 4$–$3$ and [CI] $^3P_1$–$^3P_0$ emission lines[33].

All interferometric images presented in this work are generated using CASA package version 6.3.0.

### Nested posterior sampling

To obtain a statistically robust detection of the potential SZ signal in the direction of the Spiderweb complex, we use the approach already used in refs. [16,39–41] in the context of ALMA+ACA studies of the SZ signal. In brief, we perform a visibility-space analysis, which allows for exactly accounting for the non-uniform radio-interferometric transfer function, as well as taking advantage of the Gaussian properties of noise in the native Fourier space. Any extended model component is created in image space, taking into account the proper frequency scaling and primary-beam attenuation for any fields and spectral windows used in the analysis, and is then projected onto the visibility points by means of a non-uniform fast Fourier transform algorithm based on convolutional gridding (as implemented in the finufft library[42]). Instead, in Fourier space, point-like sources are trivially represented by a constant function with amplitude equal to the source flux corrected for the primary-beam attenuation at the source position and with a phase term defined by the offset between the point source and the phase centre of the observations. To allow for Bayesian model selection and averaging, the sampling of the posterior distribution is performed by means of the nested sampling algorithm[43,44]. We specifically exploit the implementation provided in the dynesty (ref. [45]) package, which allows for robustly extending the sampling problem to moderate-dimensional and high-dimensional models. For any further details on the model reconstruction, we refer to the discussion provided by Di Mascolo et al.[16,39].

Obtaining a thorough description of the small-scale, complex morphology of the extended radio signal from the Spiderweb galaxy would require performing a pixel-level model inference (see 'Sparse imaging' section below), resulting in a posterior probability function with extreme dimensionality. Nested sampling techniques generally show better performances than Monte Carlo Markov chain algorithms in the case of moderate-dimensional problems (mitigating the impacts of the so-called 'curse of dimensionality'[44,46]). Still, sampling from high-dimensional posterior distributions may easily become computationally intractable owing to the complexity of estimating high-dimensional marginal likelihoods. Therefore, we decide to perform a first modelling run only on a large-scale subset of the available data. The giant Lyα nebula observed to surround the Spiderweb galaxy is in fact expected to be confined within a diffuse halo of hot intracluster gas[8,9,47–49]. In turn, the SZ footprint of potential intracluster gas within the Spiderweb protocluster should be expected to extend over characteristic scales ≳10″. We therefore introduce an upper cut in the visibility space at a $uv$ distance of 65 kλ, whose corresponding angular scale is roughly twice the transverse width of the jet structure (that is, the size measured along the direction perpendicular to the jet direction) observed when imaging only the high-resolution ALMA data from the C43-3 and C43-6 observations. Such a choice makes the jet signal spatially resolved only along the jet axis and allows for describing this as a limited collection of point-like sources. As a result, the number of parameters required to model the observations remains limited and the analysis computationally manageable, while allowing for using nested sampling to obtain statistically meaningful information on our model inference.

The selection of the specific number of compact components required to model the radio source was performed by means of Bayesian model selection. In particular, we consider as the most favoured model set the one for which the introduction of an extra point-like term would have caused a degradation or only a marginal improvement in the log-evidence[50] (that is, $(\log Z_{n+1} - \log Z_n) \lesssim 0.50$, in which $n$ denotes the number of model components). For all of the components, as already described above, we assume the spatial morphology to be described by a Dirac $\delta$ function (that is, a constant function in Fourier space with non-null phase term) and the source fluxes by a power-law spectral scaling. The priors for all the parameters—right ascension, declination, flux and spectral index—are described by uniform probability distributions. In particular, the right ascension and declination are allowed to vary within the area of $r \simeq 1'$ enclosed within the first null of the antenna pattern for the Band 3 ALMA data at the highest available frequency (that is, 107.5 GHz). To avoid label switching and force mode identifiability, we impose a further ordering prescription[51] to the right ascension

parameters. The flux and spectral index parameters are instead allowed to vary within uniform prior probability distributions. In particular, we constrain the source fluxes to be non-negative and assume an upper limit of 10 mJy on each amplitude, around an order of magnitude larger than the emission peak in the ALMA map (1.54 mJy; see top panel of Extended Data Fig. 1). For the spectral indices, we consider a range [−5, 10], arbitrarily wide and set to cover both the cases of power spectra with negative and positive slopes, consistent with synchrotron-like and dust-like spectral properties, respectively. The prior limits are intentionally set to extend well beyond the values expected for such cases, to avoid overconstraining of the source spectral properties while allowing for a quick diagnostics of the actual constraining power of the available data with respect to spectral information.

To describe the pressure distribution of the ICM, we instead use a generalized Navarro–Frenk–White (gNFW) profile[52], widely shown to provide an accurate description of the average pressure distribution of the intracluster gas. In particular, we use different gNFW formulations from the literature. A summary is provided below.

–The universal profile (hereafter, A10 UP) by Arnaud et al.[28] derived from the reconstruction of the pressure distribution in galaxy clusters from the REXCESS[53] sample and the equivalent model obtained from the subset of systems with clear evidence of a disturbed morphology (A10 MD). Although this profile is calibrated on massive local systems ($10^{14} M_\odot < M_{500} < 10^{15} M_\odot$, $z < 0.2$), it is the base model used for the mass reconstruction in large-scale SZ surveys and allows for a straightforward comparison with the literature.

–The pressure model[54] (M14 UP) obtained from the X-ray analysis of a high-redshift ($0.6 < z < 1.2$) subsample of galaxy clusters detected by the South Pole Telescope[55–57] (SPT). As for the previous case, we further consider the pressure profile (M14 NCC) reconstructed by excluding all the systems with clear evidence for the presence of a cool core, generally indicative of a more relaxed dynamical state. To our knowledge, this model represents the highest-redshift, observationally motivated pressure profile available at present, in turn potentially providing a better description of the pressure distribution in the Spiderweb system than the A10 models.

–The median pressure profiles reconstructed from the OverWhelmingly Large Simulations (OWLS) suite of cosmological hydrodynamical simulations, cosmo-OWLS[29], and considering different prescriptions for the physical model of AGN-driven heating of the intracluster gas. In particular, we consider the OWLS[58] reference model (L15 REF), as well as the two AGN models (L15 8.0 and L15 8.5; we refer to Le Brun et al.[29] for details). These models present two main advantages. First, they are built on simulated halos whose mass and redshift ranges ($2 \times 10^{12} M_\odot \lesssim M_{500} \lesssim 3 \times 10^{15} M_\odot$, $0 < z < 3$) broadly overlap with the properties of the Spiderweb protocluster. Second, at $z \simeq 2$, the Spiderweb complex sits in a transitional phase for AGN feedback[59,60] and the different flavours of the cosmo-OWLS pressure model allow for directly using the SZ effect to explore different AGN scenarios.

–The mass-dependent and redshift-dependent extended pressure model[61] (G17 EXT) calibrated on simulated galaxy clusters from the set of Magneticum Pathfinder hydrodynamical simulations (http://www.magneticum.org/). Although this profile is computed on massive galaxy clusters ($M_{500} > 1.4 \times 10^{14} M_\odot$), it provides an explicit model for taking into account the departure from universality and self-similarity as a function of mass and redshift.

In all cases, the free parameters defining the gNFW profiles are the plane-of-sky coordinates of the model centroid and the mass parameter $M_{500}$. As for the radio model, the right ascension and declination parameters are prescribed to vary within the region encompassed by the first null of the Band 3 ALMA primary beam. For the mass $M_{500}$, we consider a log-uniform distribution over the range [$10^{12} M_\odot$, $10^{15} M_\odot$], to facilitate the posterior exploration over such a wide range of order of magnitudes.

We also tried fitting the cool-core versions of the A10 and M14 profiles above but found these to be systematically disfavoured ($\Delta \log Z \lesssim 15.5$) in comparison with the listed models. This is, however, not surprising, as the presence of a well-formed cool core would be hardly consistent with the inherently disturbed nature of a protocluster complex.

Finally, we account for any potential systematics with data calibration by considering a scaling parameter for each of the measurement sets used in our analysis. For these, we assume normal prior distributions, with unitary central value and standard deviation of 5%, as reported in the ALMA Technical Handbook for the considered observing cycles.

**Results.** With regards to the extended signal from the Spiderweb galaxy, the criterion introduced above for the selection of the number of point-like components supports the case for a total of eight distinct components over the entire search area (we summarize the key information on the results of the point-like modelling in Extended Data Table 1). In particular, two components (ID1 and ID2) are found to be spatially consistent with the position of known protocluster members[25,34,35,49,62–64]—that is, ERO 284 (ref. [12]) and HAE 229 (refs. [65,66]), respectively—located around 250 kpc west of the Spiderweb galaxy. The remaining components instead uniquely describe the radio emission associated with the Spiderweb galaxy, with one point-like component (ID8) specifically corresponding to the bright lobe of the eastern radio jet and another (ID6) being nearly coincident with the Spiderweb galaxy itself (Extended Data Fig. 1). All the components describing the extended signal exhibit a negative spectral index, consistent with the synchrotron origin of the emission. The best-fit estimates highlight a spatial variation consistent with what is observed at lower frequencies in the Karl G. Jansky Very Large Array (VLA) data[6–9], which show that the spectrum of the eastern lobe is, on average, steeper than that of the Spiderweb galaxy and the western jet. The specific values are also in rough agreement with the results from the VLA analyses. A one-to-one comparison is, however, not practicable, owing to the inherent high-frequency spectral steepening induced by radiative losses and the different modelling approaches. The two offset sources, on the other hand, are both characterized by positive spectral indices. Such a trend is possibly indicating a dominant contribution from thermal dust emission already at about 100 GHz and is in agreement with the potential presence of massive dust reservoirs in the galaxies, as already verified for HAE 229 (refs. [65,66]).

A summary of the inferred parameters for the different SZ models is instead provided in Extended Data Table 2. Despite the more or less substantial differences in the reconstructed parameters, all the adopted pressure profiles resulted in statistically consistent SZ models, not allowing a statistically motivated selection of a specific description. All the assumed models, with exception of the L15 8.0 and L15 8.5 cases, provide mass estimates $M_{500} \simeq 3 \times 10^{13} M_\odot$. In fact, we note that the L15 8.5 profile from Le Brun et al.[29] results in a mass of about $8 \times 10^{13} M_\odot$, consistent with many of the dynamical estimates reported in the literature[11,13,15] for the Spiderweb protocluster. Considering that the profile is based on simulations with the prescription for an intense heating of the ICM owing to AGN feedback, the result might hint to a crucial role of the active core of the Spiderweb galaxy in shaping the intracluster/circumgalactic medium. Nevertheless, this model is statistically equivalent to many others in our sample, limiting the statistical relevance of the above considerations. For the same reason, any attempts of performing a comparison with past ICM studies of similarly high-redshift systems—for example, XLSSC 122 (ref. [67]) or Cl J1449+0856 (ref. [68]), the clusters at the highest redshift known so far with a direct SZ measurement—would not be statistically meaningful.

Finally, we note that, in our analyses, all the scaling parameters are found to be broadly consistent with unity. In particular, the scaling factors are measured to be equal to $1.04^{+0.04}_{-0.04}$ and $1.01^{+0.03}_{-0.04}$ for the Band

3 ACA and ALMA observations, respectively, whereas—in the case of the Band 4 data—we find parameters of $1.00^{+0.06}_{-0.07}$ for ACA and $1.03^{+0.04}_{-0.04}$ for ALMA (the reported values are given by the Bayesian model averages for all models considered in this work; see the 'Dependence of the SZ significance on the number of point-like components' section below for a discussion).

**Comparison with masses from previous studies.** Performing a proper comparison of the SZ-derived mass estimates (Extended Data Table 2) with independent measurements from the literature is non-trivial. The dynamical masses for the Spiderweb protocluster are in fact based on velocity-dispersion estimates that might trace specific, yet not well-identified substructures and that span almost an order of magnitude: from 204 km s$^{-1}$ for one peak in the velocity distribution of Ly$\alpha$ emitters in the Spiderweb field[11] and up to 1,360 km s$^{-1}$ as measured for all satellites within 60 kpc from the Spiderweb galaxy[14]. Nevertheless, the SZ-based mass estimates presented in this work are much lower (a factor of about 2–4, depending on the model) than the dynamical values[11,14,15] found in the literature for the mass of the whole protocluster structure. In fact, the same studies reported evidence for a complex velocity structure within the central region of the Spiderweb complex, hinting at the possibility for the system to be experiencing a major merger and still accreting large amounts of material from surrounding filaments. The fact that the Spiderweb system is embedded in a large-scale filamentary structure was confirmed by wide-field CO $J = 1-0$ mapping[35]. As such, even the very core of the Spiderweb protocluster might not be fully virialized. Accordingly, the integrated SZ signal $Y_{SZ} = Y_{SZ}$ (<5$r_{500}$) we measure from the ALMA+ACA observations— $Y^{ALMA}_{SZ} = (1.68^{+0.35}_{-0.32}) \times 10^{-6}$ Mpc$^2$ (see Extended Data Table 2)—is, for instance, a factor of 3.5 times lower than that expected, taking the mass inferred from the velocity-dispersion measurement $\sigma_v \simeq 683$ km s$^{-1}$ reported by Shimakawa et al.[15] for the galaxies within the 0.53-Mpc region surrounding the Spiderweb galaxy.

This large difference between the expected SZ signal from velocity-dispersion measurements and the observed ALMA+ACA SZ integrated flux could be the result of a scenario in which the measured SZ signal is dominated by the contribution from the most prominent (sub)halo. In fact, the integrated SZ flux $Y_{SZ}$ scales steeply as a function of mass $M$, meaning that the SZ signal from a single halo would be larger than the sum of the SZ flux from a complex system of subhalos whose masses amount overall to the same value $M$. Under the assumptions that the Spiderweb protocluster is composed of several interacting substructures and the measured line-of-sight velocity dispersion $\sigma_v$ is providing an unbiased estimate of the total mass of the system, we can exploit the $Y_{SZ}$–$M$ relation to obtain an estimate of the number $n_{halos}$ of subhalos populating the Spiderweb complex.

First, the dispersion $\sigma_v$ is converted into a dynamical mass estimate using the scaling relation calibrated by Saro et al.[69] on a mock galaxy catalogue from the Millennium Simulation[70]. This is rescaled to $M_{500}$ assuming the conversion relation in Ragagnin et al.[71] between masses at different overdensities. We then iterate over $n_{halos}$ and compute the integrated SZ flux expected for each set of subhalos. For the sake of simplicity, we consider all the $n_{halos}$ subcomponents to have equal mass $M^{\sigma_v}_{500}/n_{halos}$. Both the masses from our SZ analysis $M^{ALMA}_{500}$ and from the velocity-dispersion measurement $M^{\sigma_v}_{500}/n_{halos}$ are then used to obtain a measurement of the spherically integrated SZ flux $Y_{SZ}$ by means of numerical integration of the pressure profiles over a volume of radius 5$r_{500}$. In our calculations, we consider the universal formulation by Arnaud et al.[28] to describe the pressure distribution of the intracluster electrons. Finally, to derive the number of equal-mass subhalos within the Spiderweb protocluster whose individual SZ fluxes would match the measured value $Y^{ALMA}_{SZ} = Y_{SZ}(M^{ALMA}_{500})$, we simply estimate the value $n_{halos}$ for which the equality $Y_{SZ}(M^{ALMA}_{500}) = Y_{SZ}(M^{\sigma_v}_{500}/n_{halos})$ is satisfied (see Extended Data Fig. 2). We note that adopting an underlying pressure model different from the A10 profile used for producing

Extended Data Fig. 2 is not causing a relevant variation in the total number of subcomponents $n_{halos}$, which are overall constrained to range between two to a maximum of four (except for the L15 8.5 case, resulting in $n_{halos} \simeq 1$). We further note that we made several attempts in using a physically motivated subhalo mass function (for example, the theoretical model provided by Giocoli et al.[72]) instead of our simple equal-mass distribution. However, we found that removing the strong constraint of having subhalos with the same masses makes the derivation of $n_{halos}$ or of the subhalo mass parameters unconstrained, resulting in unstable and heavily degenerate results. We note, in any case, that all these considerations are derived a posteriori of the SZ modelling and, thus, do not affect the significance of the reported SZ detection.

This emerging multihalo picture is also consistent with the identification reported in the literature[14,20] of double peaks in the velocity distribution. Nevertheless, we do not identify in the posterior distribution for the SZ model any notable secondary peaks that would be indicative of several pressure components in the ICM[40,41] in the direct surroundings of the Spiderweb galaxy (see however the discussion below on the results of a multicomponent analysis). This might be caused by the chance line-of-sight alignment of separate halos, but neither the spectroscopic information on the protocluster members nor the ICM constraints allow us to disentangle any distinct contributions from superimposed substructures. Similarly, any subhalos with similar masses would also be characterized by comparable pressure distributions, in turn causing their SZ signal to be barely distinguishable. Overall, the above result suggests that the SZ effect is tracing a minor portion of the larger Spiderweb structure in which the ICM has started building up and pressurizing, whereas the rest of system, extending over scales of tens of Mpc (refs. [11,12,20,25,35,49,65]) and tracing the region encompassed by the turnaround radius of the overdensity, has yet to undergo virialization. Any SZ signals associated with further subhalos or more extended structure are not constrained by the observations, which have limited sensitivity and may suffer from large-scale interferometric filtering.

**Multiple SZ components.** The synchronous multiellipsoidal sampling[73] typical of the main nested sampling algorithms—and, in particular, of dynesty (refs. [45,74]), the library used for our analysis—would naturally break into separate posterior modes in the presence of several peaks in the posterior density function. This would be the case, for instance, in the presence of several halos, resulting in distinct SZ components (as reported in, for example, refs. [40,41] in the case of merging cluster systems). However, as mentioned above, we do not find such evidence in the posterior probability distribution for any of the main modelling runs presented in this work. Nevertheless, we tested for the potential presence of any extra SZ features by performing a multicomponent analysis. In particular, we consider the same model description as in the single-halo case above but introduce further SZ terms. As in the case of the radio-source modelling, to avoid label switching, we introduce an ordering condition on the centroid coordinates of the SZ components. This is applied first to the right ascension parameters and then to the declination direction, to test against any bias potentially introduced by the specific prior choice. Nevertheless, the results are found to be consistent between the two modelling sets.

Independently of the model used to describe the underlying pressure distribution, the sampler identifies a secondary SZ feature 27.3$^{+2.4''}_{-3.8}$ southeast of the Spiderweb galaxy, corresponding to approximately 226 kpc at the protocluster redshift. This falls right at the $r_{500}$ boundary of the main SZ component, implying that this secondary feature, if real, would be associated with a halo distinct from the one in which the Spiderweb galaxy is embedded. The actual existence of such a structure however cannot be firmly assessed. The images produced with the high-resolution algorithm (see the 'Sparse imaging' section below) or after subtracting the radio-source model from the low-resolution set

do not provide any clear evidence for any off-centre SZ structure. A lack of spatial correspondence is also noted with respect to the protocluster members, as the secondary SZ component cannot be clearly associated with any specific concentration of member galaxies, indicative of a distinct collapsed halo. The absence of a correspondence with protocluster galaxies further limits (if not excludes) the chances for the SZ component to be associated with the secondary velocity peak mentioned above. Above all this, the Bayesian evidence of the model comprising two SZ components improves on the one-component case by a factor of only $(\log Z_{n_{sz}=2} - \log Z_{n_{sz}=1}) \lesssim 1.84$, corresponding to an effective significance of $\sigma_{eff} \lesssim 1.90$. As such, the ALMA+ACA data available at present are not able to support the unequivocal, statistically significant identification of a secondary pressure component.

In fact, increasing the flexibility of the SZ model beyond the two-halo scenario does not provide any effective improvements in the overall reconstruction. In particular, the inclusion of a third component induces the Bayesian evidence to degrade, with a reduction with respect to the two-component case of $(\log Z_{n_{sz}=3} - \log Z_{n_{sz}=2}) \lesssim -1.67$ and a limited improvement with respect to the reference single-halo model $(\log Z_{n_{sz}=3} - \log Z_{n_{sz}=1}) \lesssim 0.17$ (which converts to $\sigma_{eff} \lesssim 0.60$). At the same time, we observe that allowing for an ellipsoidal pressure distribution provides a marked improvement in the significance of the model $(\sigma_{eff} \lesssim 4.41)$. However, this concurrently makes the sampling converge to a hardly physical solution, with a plane-of-sky eccentricity $\varepsilon = 0.96^{+0.02}_{-0.04}$—that is, corresponding to a plane-of-sky minor axis being only 4% of the respective major axis. This is mainly a consequence of the strong degeneracy between the mass (that is, the parameter controlling the overall amplitude and scale radius of the SZ signal) and the line-of-sight extent of the ICM distribution, governed by the eccentricity parameter $\varepsilon$. In fact, SZ data alone cannot provide information on the line-of-sight distribution of the optically thin ICM. We thus have to force the line-of-sight scale radius to be equal to the geometric mean of the major and minor axes of the three-dimensional ellipsoid, assumed for simplicity to lie on the plane of the sky.

Overall, the main consequence for the main SZ detection with data available at present is that a single spherically symmetric halo is sufficient to provide a statistically exhaustive description of the SZ footprint of the Spiderweb protocluster. The result of the elliptical modelling can only be interpreted as a marginal indication of an underlying morphological complexity, without however providing a conclusive and meaningful answer on the spatial properties of the SZ signal. At the same time, as demonstrated for the radio-source model, any resolvable asymmetry should be naturally traced by an ordered multicomponent model. Gaining a better understanding of the morphological properties of the forming ICM would require achieving improved quality from the observational side, in terms of both sensitivity and frequency coverage. Most importantly, though, the results reported above suggest that the robust identification of the SZ signal already with the simple spherical model provides only a lower limit to the actual significance of the detection and that this could only be enhanced when including in our models the description for any irregular and asymmetric features.

**Dependence of the SZ significance on the number of point-like components.** First, we note that we do not observe a substantial variation in the fluxes of the compact components between the modelling runs with and without a SZ component. This provides a straightforward test of the robustness of our model reconstruction, in particular assuring against being driven in the SZ identification by the oversubtraction of the radio source. To properly assess whether the significance of the SZ signal is, however, dependent on the specific assumption on the number of point-like components, we rerun the SZ modelling for the entire sample of model setups considered for finding the optimal set of compact sources. A summary of such a test for our reference model (A10 UP) is provided in Extended Data Fig. 3. We note that we consider

here only the case $n > 4$, as this is found to represent the minimal condition for observing a sensible improvement in the image-space residuals and for the sampler not to suffer from slow convergence.

The first outcome is that increasing the number of point-like components beyond our reference model ($n = 8$) induces a drop in the Bayesian evidence, thus not justifying any further extension of the radio source to $n \geq 9$ (second panel). This implies that, despite all the parameters remaining practically unvaried beyond the optimal set with $n = 8$, increasing the number of point-like sources to $n \geq 9$ makes the modelling incur data overfitting. On the other hand, before $n = 6$, the models exhibit a rapid increase in the overall significance in comparison with the null case, that is, the data-only run with no model components except for the cross-data calibration parameters (first panel). Concurrently, for $n \leq 5$, the right ascension and declination of the SZ centroid (fourth and fifth panels) are observed to roughly collapse on the values inferred for the secondary SZ feature identified at low significance in the multicomponent run discussed above. Corresponding to a region with low primary-beam amplitude ($\lesssim 0.50$, depending on the specific array and band), this is compensated by an abrupt increase in $M_{500}$ when moving to $n \leq 5$ (sixth panel), in turn resulting in a more extended (and thus more severely filtered) SZ signal. Despite being naively favoured by statistical reasoning on $\sigma_{eff}^{SZ}$ (third panel), we note that the overall effective significance of the SZ models for $n \leq 5$ is, however, substantially lower than the stable $n \geq 6$ cases (first panel), thus limiting the validity of this reconstruction. Further, the cases $n \leq 5$ correspond to radio-source models that substantially underfit the data and fail in describing the complex morphology of the extended emission from the Spiderweb galaxy (for this, we refer to Extended Data Fig. 4). As such, the identification of such an offset SZ feature cannot be reliably associated with the actual presence of any physical component and might be induced by spurious systematic features in the visibility data.

Despite overall supporting the effectiveness of our modelling and choice for the reference model, the above results clearly highlight a marginal level of variance introduced by the different assumptions on the number of compact sources. To take this into account and to limit any bias consequent to the choice of a specific model as reference, we thus decide to use Bayesian model averaging[75,76] to generate model-marginalized profiles when comparing our reconstruction with observations (see Fig. 1). In practice, we computed cross-model posterior probability distributions by applying a weighted average to the original model posteriors, with the weight of each sampling point set equal to the respective Bayesian evidence. To fully account for any degeneracies between different model parameters, we generate radio and SZ $uv$ models for each posterior sample and each pressure model, and then apply the Bayesian model averaging reduction to the resulting collection of $uv$ profiles.

**Systematics in the Bayesian analysis.** We note that, because the dynamical mass estimates are computed assuming virial equilibrium for the entire structure, their values represent upper limits to the real mass of the Spiderweb protocluster. On the other hand, our SZ-derived mass estimates might be affected by non-trivial systematic biases associated with the assumptions used in our modelling.

First, the conversion between SZ signal and total mass is derived under hydrostatic equilibrium considerations. Given the disturbed nature of a galaxy protocluster, we can instead expect notable non-thermal contributions to the overall pressure support to the ICM—ranging from turbulent motion to dynamical effects caused by recent or continuing merger events. At the same time, hydrodynamical simulations[77–79] show that the growth of the mass of a system resulting from a merger event does not correspond to a concomitant increase of the thermal SZ signal, owing to the temporal offset between mass evolution and gas thermalization. If the Spiderweb protocluster is observed while experiencing a merging process, the intrinsic SZ signal

would thus be lower than expected from standard mass-observable scaling relations.

Second, the reconstruction of the SZ signal relies on the assumption of self-similarity across cosmic time of the halo properties. Although this is observed[80] to hold on large scales for galaxy clusters even up to $z \simeq 1.9$, we still lack an exhaustive description of the average thermodynamic properties for the ICM within protocluster complexes, as well as any observational information for the potential self-similar appearance of protocluster halos with the ones in their $z \lesssim 2$ descendants. Also, for the sake of computational feasibility, we rely on the relatively strong assumption of spherically symmetric distribution for the electron pressure within the ICM. Nevertheless, the overall depth of the available data limits the possibility of obtaining better constraints on the SZ effect from the Spiderweb complex or achieving an improved separation of the signals from the radio source and the underlying SZ effect able to highlight any diffuse ICM halo with low surface brightness. Similarly, asymmetries in the morphology of the pressure distribution and, therefore, in the resulting SZ distribution, as well as to any secondary ICM components populating the Spiderweb complex, cannot be firmly assessed (see, however, 'Multiple SZ components' above for a discussion).

Third, given the limited information across the millimetre/submillimetre window with sufficient sensitivity to constrain the SZ spectrum, we are not able to disentangle any contribution to the overall SZ signal in the direction of the Spiderweb protocluster from the kinetic term. Similarly, we cannot constrain the relativistic correction to the SZ spectrum. Such effect is however generally subdominant at the virial temperature expected for such a low-mass system (for example, the relativistic correction to the thermal SZ effect at 2 keV is on the order of 1.2%; we refer to Mroczkowski et al.[3] for a review of the different contributions to the SZ effect). Therefore, we assume the measured SZ signal to be entirely caused by its non-relativistic thermal component.

## Sparse imaging

The standard tools available at present for performing radio-interferometric imaging and deconvolution are not optimally suited to the joint reconstruction of the signal from an extended radio source superimposed over a diffuse SZ decrement. Cross-contamination may in fact cause both an underestimation of the flux of the former when using CLEAN-like algorithms[81], and—at the same time—introduce a notable suppression of the underlying SZ footprint. Also, common approaches exploiting scale separation between the signature from the radio sources and the ICM in galaxy clusters[82,83] would not be able to provide a robust characterization of the two signals, as their characteristic scales exhibit a broad overlap in the case of the Spiderweb protocluster.

Building on the extensive literature on compressed sensing[84–86] and sparsity-based component separation[87,88] and imaging[89–92], we thus developed an algorithm for taking full advantage of both the different and, in the case of the SZ effect, well-constrained spectral behaviour of the measured signals, as well as the information on the different spatial-correlation properties. In particular, we assume the total surface brightness $I_\nu$ in a given direction on the plane of sky $(x, y)$ and at a given frequency $\nu$ to be described as

$$I_\nu(x, y) = I_{RS}(x, y) \cdot (\nu/\nu_0)^{\alpha(x,y)} + I_{SZ}(x, y) \cdot g_{SZ}(\nu) \qquad (1)$$

Here $I_{RS}(x, y)$ is the surface brightness of the radio source computed at the reference frequency $\nu_0$, whereas $\alpha(x, y)$ is the corresponding spatially varying spectral index. The terms $I_{SZ}(x, y)$ and $g_{SZ}(\nu)$ denote, instead, the amplitude in units of Compton $y$ and spectral scaling for the thermal SZ effect[2,3]. Because, as already discussed above, we are not including in our SZ model any corrections owing to relativistic terms, the spectral SZ scaling is determined solely by the frequency

and therefore does not contribute to the overall model through any specific free parameter. To achieve a high-fidelity reconstruction of the complex morphology of the structure of the radio jet from the Spiderweb galaxy, we model $I_{RS}(x, y)$ and $\alpha(x, y)$ as a collection of pixels with angular scale matched to the longest baseline according to the Nyquist sampling theorem (given the maximum $uv$ distance of $u_{max} \simeq 1.3$ M$\lambda$, we set the pixel scale equal to $0.5/u_{max} \simeq 0.07''$). On the other hand, the signal-to-noise ratio of the SZ effect from the Spiderweb complex is not high enough to allow for adopting an analogous approach to constrain the SZ decrement. To exploit the expected large-scale correlation, we hence describe $I_{SZ}(x, y)$ through an isothermal, circular $\beta$ model[93], with free centroid coordinates, amplitude normalization and core radius, whereas we arbitrarily fix $\beta$ to a value of 1 (commonly used in ICM studies and adopted, for example, by the SPT collaboration as the base for their matched-filter cluster template[56,57,94]). We tested against potential biases introduced by the specific choice of $\beta$ but did not observe a dependence of the solution on this parameter. Clearly, this model is less refined than the gNFW profile broadly used in the literature and in the low-resolution analysis presented above. Nevertheless, here, we are mostly interested in capturing the average properties of the faint SZ distribution, while providing a good estimate of the base level on top of which to measure the surface brightness of the radio structure and avoid its undersubtraction when imaging the SZ effect.

Using a non-parametric approach for reconstructing the radio jet has the clear advantage of allowing for constraining the signal on the finest scales accessible in the ALMA+ACA measurements. However, this causes the model to comprise $O(10^4)$ free parameters, making the inference problem intractable through posterior sampling methods (for example, Markov chain Monte Carlo, nested sampling). At the same time, solving simultaneously for all the free parameters in the model of equation (1) through an optimization scheme would imply facing the shortcomings of a non-convex likelihood function. We hence use an iterative, block-coordinate optimization process[95] with proximal point update[96–98]. In this scheme, we alternatively solve at a time only for one set of parameters—namely, the radio-source amplitude, its spatially varying spectral index and the SZ centroid coordinates, normalization and core radius—through a box-constrained limited-memory Broyden–Fletcher–Goldfarb–Shanno algorithm[99,100]. We use the implementation provided in the SCIPY[101] package and assume all the standard values for controlling the convergence of the algorithm. The selection of the specific coordinate set to solve for is performed by means of the Gauss–Southwell rule[102,103], which introduces an ordering in the optimization blocks based on the norm of the Jacobian of the corresponding likelihood function.

When modelling the radio-source amplitude, we consider a LASSO-like regression[104] step to promote sparsity,

$$L_{RS} = -\frac{1}{2}\chi^2_{RS} + \lambda_{\ell_1} \|I_{RS}\|_{\ell_1} \qquad (2)$$

in which $\chi^2_{RS}$ is computed when fixing all the model parameters but $I_{RS}$, whereas the constant $\lambda_{\ell_1}$ controls the strength of the $\ell_1$-norm regularization. To mitigate the bias and amplitude truncation[105] introduced by the $\ell_1$ norm, we further iterate over the full block-coordinate loop after reweighting the regularization constant according to the previous solution for the radio source. At every main step $k$ of the optimization algorithm, we scale the regularization hyperparameter as

$$\lambda_{\ell_1}^{(k)} = \frac{\epsilon^{(k-1)}}{\epsilon^{(k-1)} + \left\|I_{RS}^{(k-1)}\right\|_{\ell_1}} \lambda_{\ell_1}^{(k-1)}. \qquad (3)$$

in which $\epsilon^{(k)}$ is an auxiliary coefficient aimed at ensuring numerical stability. We assume this to decrease with the number of reweighting iterations, $\epsilon^{(k)} = \max(\epsilon^{(0)} 10^{-k}, \sigma_{pix})$, with $\epsilon^{(0)}$ given by the standard

deviation per pixel of the signal as measured directly from the dirty image and $\sigma$ is the RMS of the image-space noise per pixel[105]. Similarly, we can express the initial value $\lambda_{\ell_1}^{(0)}$ in units of $\sigma_{pix}$ as[106] $\lambda_{\ell_1}^{(0)} = n\sigma_{pix}\Sigma_i w_i$, in which $w_i$ are the weights of the individual visibilities and $n$ is a user-defined quantity controlling the overall depth of the modelling process (that is, the optimization algorithm neglects any image-space features with significance lower than about $n\sigma_{pix}$). Each block-coordinate optimization loop is then run until the relative variation in the value of the likelihood function between two full sets of updates falls below an arbitrary tolerance of $10^{-8}$. The same tolerance on the relative change of the likelihood function between two consecutive solutions is further used for interrupting the iteration over the reweighting steps.

For the starting values for the model parameters, we consider $I_{RS}^{(0)}$ to be null everywhere and set $\alpha_{RS}^{(0)}$ to the spectral index of $-0.70$ typical for synchrotron emission. The coordinates of the SZ distribution are initially set on the phase centre of the Band 3 ALMA observations, roughly coinciding with the Spiderweb galaxy, whereas the initial amplitude and scale radius are initially set equal to zero and to an arbitrary scale of 30″, respectively. We tested for such choices by injecting random values for each parameter block at the beginning of the optimization run and found no substantial deviations in the final results.

In our analysis, we attempted to impose an extra smoothing to the radio-source amplitude through a total squared variation regularization[107], found to guarantee, in astronomical imaging, better results than the sole $\ell_1$-norm or standard isotropic total variation. However, we observed only a minor impact on the final solution and hence decided not to include any total squared variation term in the likelihood function. Similarly, the limited significance of the available data did not allow for successfully exploiting any wavelet-based algorithms, generally demonstrated[89,106,108–110] to provide an improved performance compared with simple image-based techniques.

In our analysis, we adopt a conservative threshold $n = 4$. In our tests, a lower parameter showed improved subtraction capabilities but also deteriorated purity in the solution. As an internal cross-check, we further tested against the recovered flux of the radio source, finding a maximum-a-posteriori estimate from the sparse modelling of 2.606 mJy against the value $2.602^{+0.008}_{-0.007}$ mJy inferred using the nested sampling approach. Finally, the reconstructed source manifests a morphology that matches closely the signal measured by the VLA in X-band (Extended Data Fig. 5).

## Insights from multiwavelength information

The combination of the ALMA+ACA data with the wealth of the multiwavelength information available on the Spiderweb galaxy allows us to propose several potential scenarios that could explain the intensity and morphology of the SZ signal and the associated offset with respect to the Spiderweb galaxy itself. Although mild spatial deviations between ICM halos and central galaxies are not a surprise even in moderately disturbed systems, the identification of such a displacement can inform our observational understanding of the dynamical state of a structure (see also 'Comparison with simulated protoclusters' below). Nevertheless, the available ALMA+ACA observations are not allowing to gain any insights on the physical and thermodynamic state of the forming ICM beyond the SZ detection—for example, the fraction of the total ICM reservoir that has already thermalized, the overall virialization state, the relative contribution of large-scale accretion, mergers and AGN feedback in heating the assembling ICM, the multiscale morphology of the proto-ICM and its connection (or absence thereof) with the physical properties of protocluster galaxies. Obtaining a robust characterization of the intertwined processes determining the complex physical state of the forming ICM will thus necessarily require to further improve the observational coverage of the protocluster.

Given the inherently disturbed nature of the Spiderweb protocluster, the system could be characterized by marked contributions from the

kinetic SZ effect in the direction of intracluster substructures with non-negligible relative velocity with respect to the average bulk motion of the whole system. In particular, several studies[15,19] report that the protocluster galaxies in the region southwest of the Spiderweb galaxy exhibit a relative blueshift compared with that of the central galaxy. This was interpreted as evidence for the presence of an approaching background feature, potentially tracing a filament infalling onto the protocluster core. Assuming that such structure hosts a nascent intracluster gas, this would thus be associated with a reduction of the total SZ flux owing to the infilling of the thermal SZ signal by a kinetic SZ contribution. An approaching ICM component with the same mass as that corresponding to the observed SZ feature would require, to completely cancel out its own thermal SZ effect, a line-of-sight velocity of $v_{LoS} = -1{,}928^{+641}_{-699}$ km s$^{-1}$ when assuming the temperature estimate $T_e = 2.00^{+0.70}_{-0.40}$ keV from the Chandra X-ray data[23]. Alternatively, assuming the empirical mass–temperature scaling by Lovisari et al.[111] and the resulting temperature of $T_e = 0.85^{+0.42}_{-0.25}$ keV, the required velocity would be $v_{LoS} = -796^{+356}_{-266}$ km s$^{-1}$. Notably, the former velocity value roughly corresponds to the largest velocity reported by Miley et al.[19], whereas both estimates are consistent with the average velocity for both the Ly$\alpha$-emitting satellite members[19] or the blueshifted components[14] measured by the Spectrograph for INtegral Field Observations in the Near Infrared[112] (SINFONI) on the Very Large Telescope (VLT). The same observations further show the presence of a small overdensity of redshifted members in the region immediately northeast of the Spiderweb galaxy. Similarly, the asymmetry in the SZ effect with respect to the central galaxy could be alternatively caused by an enhancement of the total SZ decrement owing to a negative kinetic contribution.

Although a notable contribution of the kinetic SZ effect can thus explain some of the multifrequency observations, we argue that this is probably not the only process responsible for the observed asymmetry. In fact, if the kinetic SZ effect were solely responsible for the skewed SZ decrement in the direction of the Spiderweb core, the X-ray emission should not be affected. We do instead observe a marginal asymmetry between the extended halo and the Spiderweb galaxy also in the Chandra X-ray data of the Spiderweb field, from which the bulk of the extended thermal emission is reported[23,25] to have a mild shift towards the northeast with respect to the X-ray signal from the central AGN (central panel of Extended Data Fig. 6; we refer to a forthcoming, dedicated paper (M. Lepore et al., manuscript in preparation) for an extended description of the X-ray data processing). However, a further suppression of the overall X-ray signal is observed in the region southwest of the main core—analogous to the SZ case—hence suggesting a local depletion in the forming intracluster halo. Recently, Anderson et al.[8] and Carilli et al.[9] reported direct evidence for the interaction of the extended radio jets with the Spiderweb environment, which could potentially result in the inflation of cavities in the diffuse protocluster gas. As commonly observed in the cores of local cool-core clusters, the presence of buoyant bubbles would induce a reduction of both the X-ray and SZ signals over the limited regions covered by the cavities—provided that the bulk of their pressure is because of non-thermal components[113]—and thus contribute to the observed asymmetry in the ICM emission.

Although potentially important, none of the processes discussed above seem to effectively explain why the proto-ICM centroid is offset with respect to the Spiderweb galaxy (Fig. 3) and the centres of the Ly$\alpha$ halo (right panel of Extended Data Fig. 6) and of the H$\alpha$ nebula[49] in which the Spiderweb galaxy is embedded. In a relatively relaxed system, the central galaxy should in fact sit deep in the centre of the gravitational well of the protocluster complex, along with most of the gas undergoing virialization, whereas the warm gas nebula is expected[8,9,47,48] to be embedded in an extended halo of hot plasma. However, the lack of strong evidence for dominant SZ or X-ray signatures from such a large-scale halo, in combination with the numerous measurements[33,63,114] showing that the central approximately 70 kpc

in the Spiderweb complex hosts a massive reservoir of atomic and molecular gas, hints at a substantial, possibly predominant presence in the core region of warm and cold gas phases rather than ionized gas at the virial temperature. Emonts et al.[115] further report the detection of a tail in the CO $J = 1$–$0$ line emission from the surroundings of the Spiderweb galaxy, interpreted as being associated with preferential accretion of cold gas along a filament. Notably, such emission is oriented along the same direction as the asymmetry in the SZ and X-ray morphologies. At the same time, past observations provide evidence for relative motion of the Spiderweb galaxy within its own molecular halo[33,114], as well as a large-scale systematic asymmetry in the velocity distribution of the protocluster members, denoting ordered flows of galaxies towards the central galaxy[14,19]. Such motion could also naturally explain the difference observed in the radio morphology of the two opposite jets from the central radio galaxy[8,9,48] (see Fig. 3). The western lobe is in fact found to be experiencing a stronger interaction with the environment than its eastern counterpart. Although recent studies[8,9] based on VLA observations of the Spiderweb galaxy satisfactorily explained this behaviour as resulting from the inhomogeneity of the density field in the surroundings of the Spiderweb galaxy, it is not possible to exclude that the relative motion of the radio galaxy within the diffuse intracluster atmosphere is in fact contributing to the overall ram pressure enhancement. All these hints, taken together, suggest that the shift between the bulk distribution of the proto-ICM and the Spiderweb galaxy may be the net result of the merging activity experienced by the system, with a multiphase subhalo or filament infalling into the cold, compact core from the east–northeast and causing a local enhancement of the SZ and X-ray signals through shock or adiabatic compression of the intracluster gas.

It is worth mentioning that the clear displacement between the proto-ICM halo and the Spiderweb galaxy might be indicative of the subdominant role of AGN feedback in providing the energy support required to heat the assembling intracluster gas. As reported by Tozzi et al.[23], the radio galaxy might be observed in the midst of the establishment of the radio-mode feedback regime and potentially providing a subefficient heating mechanism. We can use the estimates of the non-thermal pressure and the volume of the radio-bright regions provided by Carilli et al.[9] to roughly evaluate the non-thermal energy provided by the radio jet, $E_{nth} \simeq 6 \times 10^{60}$ erg. Models of jet propagation through the ICM[116] suggest that a comparable amount should already be deposited into the shocked intracluster gas. In fact, the jet energy $E_{nth}$ is comparable with the ICM thermal energy within the approximately 60-kpc sphere encapsulating the radio jets. This means that the mechanical energy might have a non-negligible impact on the thermodynamics of the gas (and SZ flux) from the same region. However, to effectively affect the thermal energy of the ICM in a few $10^{13}\,M_{\odot}$ cluster up to $r_{500}$, the jet has to operate for an order-of-magnitude longer time. All in all, despite the broad evidence[7–9,20,25,117] for a major cross-interaction between the central AGN and the assembling environment, the available data are still not allowing for inferring a conclusive answer on the relative contribution of AGN heating (versus, for example, gravitational process as infall and major mergers) to the overall thermal energy budget of the system. An investigation of the energetics and thermodynamics of the AGN–ICM system will thus be deferred to a future study.

Finally, it is important to note that the correspondence of the dominant features and, in particular, of the lack of thermal signal to the west of the Spiderweb galaxy in both the X-ray and SZ data (see central panel of Extended Data Fig. 6) exclude that any potential undersubtraction of the signal from the extended radio source could have a dominant effect on our results (see, for example, 'Nested posterior sampling' above for a collection of tests on the model reconstruction). Nevertheless, improvements to our reconstruction would require a more extensive observational coverage, including deeper X-ray and multifrequency

SZ measurements, and higher angular resolution in particular for the latter. It is worth noting that the measured integrated Compton $Y$ parameter—$Y(<5r_{500}) = (1.68^{+0.35}_{-0.32}) \times 10^{-6}$ Mpc$^2$ —corresponds to a SZ flux density of about 0.6 mJy at 100 GHz. Such measurements are however barely in reach of current-generation subarcminute-resolution millimetre-wave facilities.

## Comparison with simulated protoclusters

The parametric modelling presented in the previous sections has the fundamental advantage of providing a straightforward and highly informative characterization of the average properties of the Spiderweb protocluster (for example, the halo mass). Nevertheless, owing to the many simplifying assumptions required to perform the model reconstruction using sensitivity-limited measurements, the analytic models used in our analysis are not able to describe the finer details of the intracluster gas. Furthermore, all the models for the ICM pressure profile available in the literature at present and explored in this work are calibrated on systems on a different mass or redshift range than the Spiderweb protocluster. In fact, we still lack an analytic description derived from observational considerations of the average thermodynamic distributions and, in particular, of the characteristic pressure profile for galaxy clusters and protocluster structures at $z \gtrsim 2$. Finally, because these models provide mean pressure profiles, they cannot capture the diversity and the possibly complex morphology of the pressure structure of the ICM in protoclusters.

The shortcomings of the parametric modelling motivates the use of hydrodynamical simulations to perform a more thorough exploration of the properties of the SZ signal measured in the ALMA+ACA observations of the Spiderweb system. We stress again that this analysis has only the scope of presenting a qualitative comparison with numerical predictions of objects evolved in a cosmological environment, without any pretence to identifying and analysing an exact numerical equivalent of the Spiderweb complex. The simulated protoclusters analysed in this work are taken from the 'Dianoga' simulation set[18]. In particular, these comprise five regions that at $z = 0$ have sizes in the range 25–40 comoving Mpc h$^{-1}$, selected from a parent 1 h$^{-1}$ Gpc$^3$ dark-matter-only cosmological volume around five massive objects (with $M_{500}$ at $z = 0$ in the range $(5.636$–$8.811) \times 10^{14}\,M_{\odot}$ h$^{-1}$). Such regions are resimulated with the $N$-body hydrodynamical code OpenGadget3 at high resolution (with dark-matter and gas-particle masses of $3.4 \times 10^7$ h$^{-1}\,M_{\odot}$ and $6.1 \times 10^6$ h$^{-1}\,M_{\odot}$, respectively). Besides hydrodynamics, which is described by an improved version[118] of the smoothed-particle hydrodynamics, these simulations include radiative cooling, star formation, chemical enrichment and energy feedback from both supernovae and AGN (we refer to Planelles et al.[17] and Bassini et al.[18] for details on previous versions of the simulations used here and to a forthcoming paper by S. Borgani et al. (manuscript in preparation) for this specific sample).

The high-resolution regions are large enough to resolve several halos, some of which—but not all—are progenitors of the main clusters selected at $z = 0$. For the purpose of this comparison, we select all systems with a mass $M_{500} \gtrsim 1.3 \times 10^{13}\,M_{\odot}$ at $z = 2.16$, for a total sample of 27 distinct objects. SZ maps are generated at the corresponding snapshot using the map-making software SMAC (ref. [119]) and projected onto the visibility plane. We note that, to simplify the comparison, the projected SZ maps are generated considering only the thermal component and they do not encompass any kinetic SZ term. Despite the obvious discrepancies associated with the specific differences in the morphological properties and dynamical states of the simulated protoclusters with respect to the Spiderweb complex, Fig. 2 shows that the mock $uv$ profiles for simulated halos with mass in the same range of the parametric estimates (approximately $3 \times 10^{13}\,M_{\odot}$) are broadly consistent with the ALMA+ACA measurements.

To further gain visual insight into the simulated protoclusters, we produce synthetic maps by projecting the simulated ALMA+ACA

visibilities back into the image space, consistent with the imaging-reconstruction process of Extended Data Fig. 6. Three out of the full sample of simulated clusters are shown in Extended Data Fig. 7, with one specifically selected as exhibiting a SZ footprint resembling that observed for the Spiderweb protocluster, one as the most massive halo in the simulated set and another as exhibiting a displacement between the SZ peak and the halo centre analogous to what is observed between the SZ signal and the central galaxy in the Spiderweb complex. We checked that, for the entire sample, the physical displacement between the dark-matter halos and the corresponding dominant galaxies were never larger than about 6 kpc, more than an order of magnitude smaller than the observed SZ-galaxy displacement in the Spiderweb protocluster. In the context of the discussion on such an offset, the halo centroid and galaxy for each of the simulated halos can thus be considered to be consistent. To introduce realistic noise and optimally reflect the noise properties of the available observations, we jackknife the original measurements by changing the sign of a randomly selected half of the visibilities. This noise-like realization is finally added to the idealized SZ data. Consistent with the $uv$ result above, we can observe an overall agreement in terms of the average amplitude of the SZ surface brightness in the reconstructed images between the SZ signal measured in the ALMA+ACA data (Extended Data Fig. 6) and the estimate for the simulated low-mass system. We note that the selection of a specific projection axis has only a minor effect on the measured SZ amplitude. Over the whole sample, the change in both the peak and the average SZ signal associated with a different projection exhibits a RMS variation of about 10%, reaching a maximum of around 30% in only three cases, corresponding to objects that manifest morphological disturbances.

In any case, what is most important in such a comparison is the possibility of gaining a sensible understanding of the effect of radio-interferometric filtering on the observed distribution of the SZ signal. In the specific case shown in the bottom row of Extended Data Fig. 7, we can observe that the displacement between the SZ distribution and the centre of the corresponding dark-matter halo (in turn roughly coincident with the position of the most massive galaxy in the structure; see above) is exaggerated by the radio-interferometric suppression of the bulk, relatively symmetric SZ signal on large scales. The net result is an enhancement of local SZ asymmetries that finally drive the definition of the observed ICM morphology. In fact, the observed offset is not indicative of an absolute displacement between the ICM and the Spiderweb galaxy but rather of a positional mismatch between the peak of the pressure distribution and the radio galaxy, typical of disturbed systems.

It is clear that, to move this comparison beyond the sole qualitative analysis and to perform a robust cross-characterization of the observed and simulated SZ signals, a broader exploration of the model parameter space in the available hydrodynamical simulations would be required. Such a detailed investigation will however be deferred to a future work.

## Data availability

This work is based on ALMA and ACA measurements (project codes: 2015.1.00851.S, 2016.2.00048.S, 2018.1.01526.S) publicly available at http://almascience.org/aq/. The Chandra observations can be obtained from the CDA (http://cxc.cfa.harvard.edu/cda/). Access to the VLT/FORS1 images is possible through http://archive.eso.org/eso_archive_main.html. The Hubble Space Telescope image shown in this work is publicly available at https://archive.stsci.edu/. At the moment, technical limitations do not allow us to guarantee safe, stable and efficient public access to the full simulation set. As such, on request, the corresponding author will provide the data products from the Dianoga cosmological hydrodynamical simulations used for the comparisons presented in this manuscript. Source data for all figures are instead provided with this paper.

## Code availability

All the analyses in this work rely on the following Python packages: numpy[120], scipy[101], matplotlib[121], aplpy[122,123]. The nested sampling is performed using dynesty[45,74]. We use astropy[124,125] to manage all the common astronomical tasks and the just-in-time compiler and automatic differentiation in jax[126] to improve the performance of the optimizations in the sparse-image reconstruction. The sparse-imaging problem is solved by means of the L-BFGS-B implementation provided in the minimize routine that is part of the scipy.optimize subpackage. To compute non-uniform fast Fourier transforms of the ALMA+ACA data, we use the Python interface to finufft[42]. Owing to limited readability, documentation and user-friendliness of their interfaces, the full analysis scripts are not yet ready for public release. We are, however, keen on collaborating with anyone willing to use the modelling tools, available from the corresponding author on request.

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

**Acknowledgements** This paper makes use of the following ALMA data: ADS/JAO. ALMA#2015.1.00851.S, ADS/JAO.ALMA#2016.2.00048.S, ADS/JAO.ALMA#2018.1.01526.S. ALMA is a partnership of ESO (representing its member states), NSF (USA) and NINS (Japan), together with NRC (Canada), MOST and ASIAA (Taiwan) and KASI (Republic of Korea), in cooperation with the Republic of Chile. The Joint ALMA Observatory is operated by ESO, AUI/NRAO and NAOJ. This work is further based in part on observations collected at the European Southern Observatory under ESO programme 63.O-0477(A). L.D.M., A.S. and M.P. are supported by the ERC-StG 'ClustersXCosmo' grant agreement 716762. A.S. is further supported by FARE-MIUR grant 'ClustersXEuclid' R165SBKTMA. A.S. and S.B. acknowledge partial financial support from the 'InDark' INFN grant. L.D.M. and P.T. acknowledge financial contribution from the agreement ASI-INAF n.2017-14-H.O. C.L.C. acknowledges support through CXC grant G09-2-1103X. H.D. acknowledges financial support from the Agencia Estatal de Investigación del Ministerio de Ciencia e Innovación (AEI-MICINN) under grant 'La evolución de los cíumulos de galaxias desde el amanecer hasta el mediodía cósmico' with reference PID2019-105776GB-I00/DOI:10.13039/ 501100011033 and acknowledges support from the ACIISI, Consejería de Economía, Conocimiento y Empleo del Gobierno de Canarias and the European Regional Development Fund (ERDF) under grant with reference PROID2020010107. M.N. acknowledges INAF-1.05.01.86.20. F.R. acknowledges support from the European Union's Horizon 2020 research and innovation programme under the Marie Skłodowska-Curie grant agreement no. 847523 'INTERACTIONS' and the Cosmic Dawn Center, which is financed by the Danish National Research Foundation under grant no. 140.

**Author contributions** L.D.M. coordinated the research project, developed the radio-interferometric imaging and modelling algorithms, performed the data analyses, planned data visualization and produced the corresponding figures, and drafted and edited the manuscript. A.S. conceived and led the ALMA+ACA proposal 2018.1.01526.S on whose data this work is based. T.M. provided substantial contributions to the interpretation of the ALMA+ACA observations and to editing the manuscript. S.B. and E.R. produced and postprocessed the 'Dianoga' suite of cosmological hydrodynamical simulations and contributed to writing the corresponding Methods section. E.C. provided guidance on the multihalo analysis and on the interpretation of the SZ results in the multiwavelength context. P.T. led the Chandra proposal and analysed and processed the corresponding observations. H.D. and K.B. contributed to shaping the original idea at the base of the ALMA+ACA observing programme on which this work is focused. H.D. further contributed to the interpretation of the radio-source modelling in relation to the observational properties of the protocluster galaxies. C.L.C. calibrated, processed and imaged the VLA data. M.N. performed the reprocessing of the optical VLT/FORS1 observations. M.G., G.M., M.P., L.P. and F.R. contributed to the interpretation of the multiphase gas properties presented in the 'Insights from multiwavelength information' section. All authors contributed substantially to discussing the results and preparing the manuscript.

**Competing interests** The authors declare no competing interests.

**Additional information**
**Correspondence and requests for materials** should be addressed to Luca Di Mascolo.

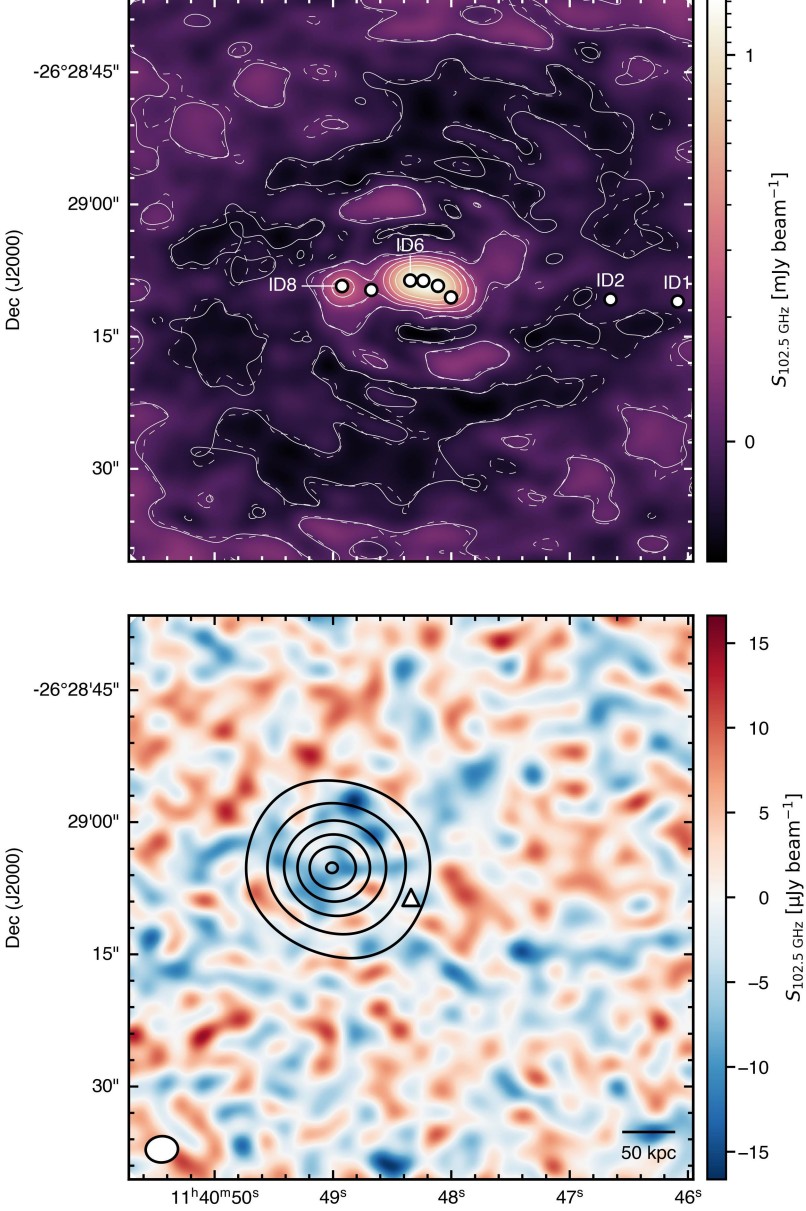

**Extended Data Fig. 1 | Images from the nested sampling reconstruction.**
Raw image (top) from the Band 3 ALMA+ACA data and residuals (bottom) after
subtraction of the marginalized model for the central radio source obtained
from the nested sampling. We stress that we are not subtracting the model
component for the SZ effect when generating the residual image but the
limited sensitivity is making any SZ features non-obvious. Because our analyses
are performed in the native visibility space, we do not perform any cleaning
step. The solid and dashed white contours in the top figure trace the data image
and the radio-source model, respectively. As evident from the comparison of
the two sets of contours, the model map is hardly distinguishable from the
data. The circles in the top panel denote the position of the point-like model
components used to described the extended signal from the Spiderweb galaxy

and the continuum emission from the western protocluster members. We label
the point-like sources that are observed to correspond to distinct features or
known astrophysical sources (see discussion in 'Results' or Extended Data
Table 1). The black contours overlaid on the residual map denote the reference
A10 UP model for the SZ signal. The SZ component manifests a clear shift
with respect to the Spiderweb galaxy, consistent with the result from the
independent sparse-imaging analysis (Extended Data Fig. 6). Consistent with
the nested sampling analysis, the images are produced after applying an upper
*uv* cut of 65 kλ to the input data. As for Extended Data Fig. 6, the triangle marks
the position of the Spiderweb galaxy. All the contour levels are arbitrary and
chosen with the sole intent of optimally marking the position of the best-fit SZ
model.

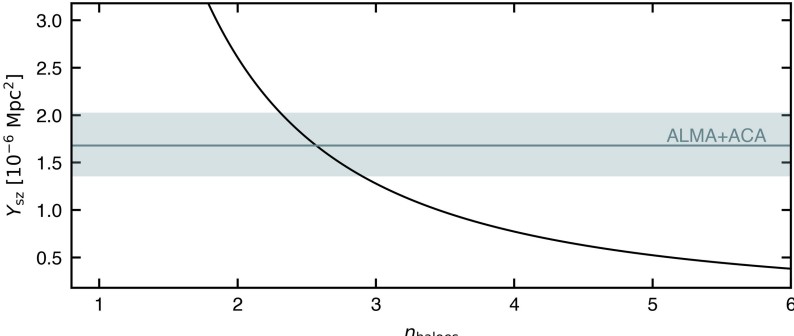

**Extended Data Fig. 2 | Integrated SZ flux as a function of the number of subhalos.** The total dynamical mass $M_{500}^{\sigma_v}$ is here computed assuming a velocity dispersion of $\sigma_v = 683\ \mathrm{km\,s^{-1}}$ as reported by Shimakawa et al.[15] for a 0.53-Mpc region around the Spiderweb galaxy. The green line and band correspond, respectively, to the median value and 68% credible interval of $Y_{SZ}^{ALMA}$ estimated from the ALMA+ACA samples for an A10 UP pressure profile. The number of equal-mass subhalos with individual SZ fluxes equal to $Y_{SZ}^{ALMA}$ and required to obtain a total mass equal to the dynamical estimates is hence found to be $n \simeq 2.57$.

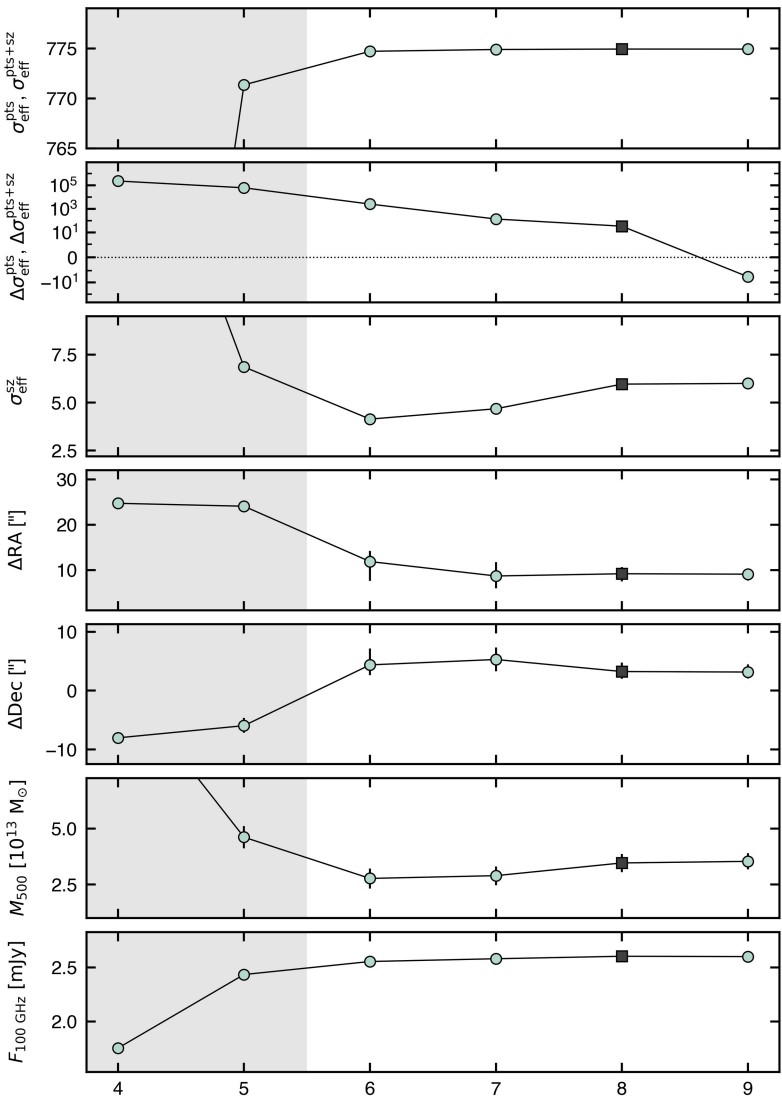

**Extended Data Fig. 3 | Impact of the number $n$ of compact sources on the model reconstruction.** From top to bottom, denoting with $\mathcal{Z}$ the Bayesian evidence for each model: effective significance $\sigma_{\text{eff}}^{\text{x}} = \text{sgn}(Z_n^{\text{x}} - Z_0^{\text{x}}) \cdot (2|Z_n^{\text{x}} - Z_0^{\text{x}}|)^{1/2}$ with respect to the null, data-only run ($n = 0$) of the models comprising only the radio-source component (x = pts) and the one including a SZ term (x = pts + sz); variation of the effective significance $\Delta\sigma_{\text{eff}}^{\text{x}} = \text{sgn}(Z_n^{\text{x}} - Z_{n-1}^{\text{x}}) \cdot (2|Z_n^{\text{x}} - Z_{n-1}^{\text{x}}|)^{1/2}$ in the radio-source-only and full models owing to the increment in the number of point-like components; effective significance $\sigma_{\text{eff}}^{\text{sz}} = \text{sgn}(Z_n^{\text{pts+sz}} - Z_n^{\text{pts}}) \cdot (2|Z_n^{\text{pts+sz}} - Z_n^{\text{pts}}|)^{1/2}$ of the SZ component with respect to the respective radio-only model; deviation in right ascension ($\Delta$RA) and declination ($\Delta$Dec) from the coordinates of the Spiderweb galaxy (11 h 40 m 48.34 s, −26° 29′ 08.55″) as for Extended Data Table 2; mass parameter $M_{500}$; total flux of the radio model. After a swift evolution in the model evidence and parameters for $n \leq 5$ (grey area), the model nearly stabilizes around the reference solution (that is, for $n = 8$, marked as a black square in all panels) before incurring a degradation of its statistical significance at $n \geq 9$ (as clearly shown in the second panel by the negative $\Delta\sigma_{\text{eff}}^{\text{pts}}$ at $n = 9$). We finally note that, despite both being plotted, the terms $\sigma_{\text{eff}}^{\text{pts+sz}}$ and $\Delta\sigma_{\text{eff}}^{\text{pts+sz}}$ are not visible because, on the scales considered in the first two panels, they practically overlap with and are indistinguishable from the respective radio-source points $\sigma_{\text{eff}}^{\text{pts}}$ and $\Delta\sigma_{\text{eff}}^{\text{pts}}$. The error bars for each point denote the amplitude of the 68% (about 1$\sigma$) credible interval for the corresponding posterior distribution.

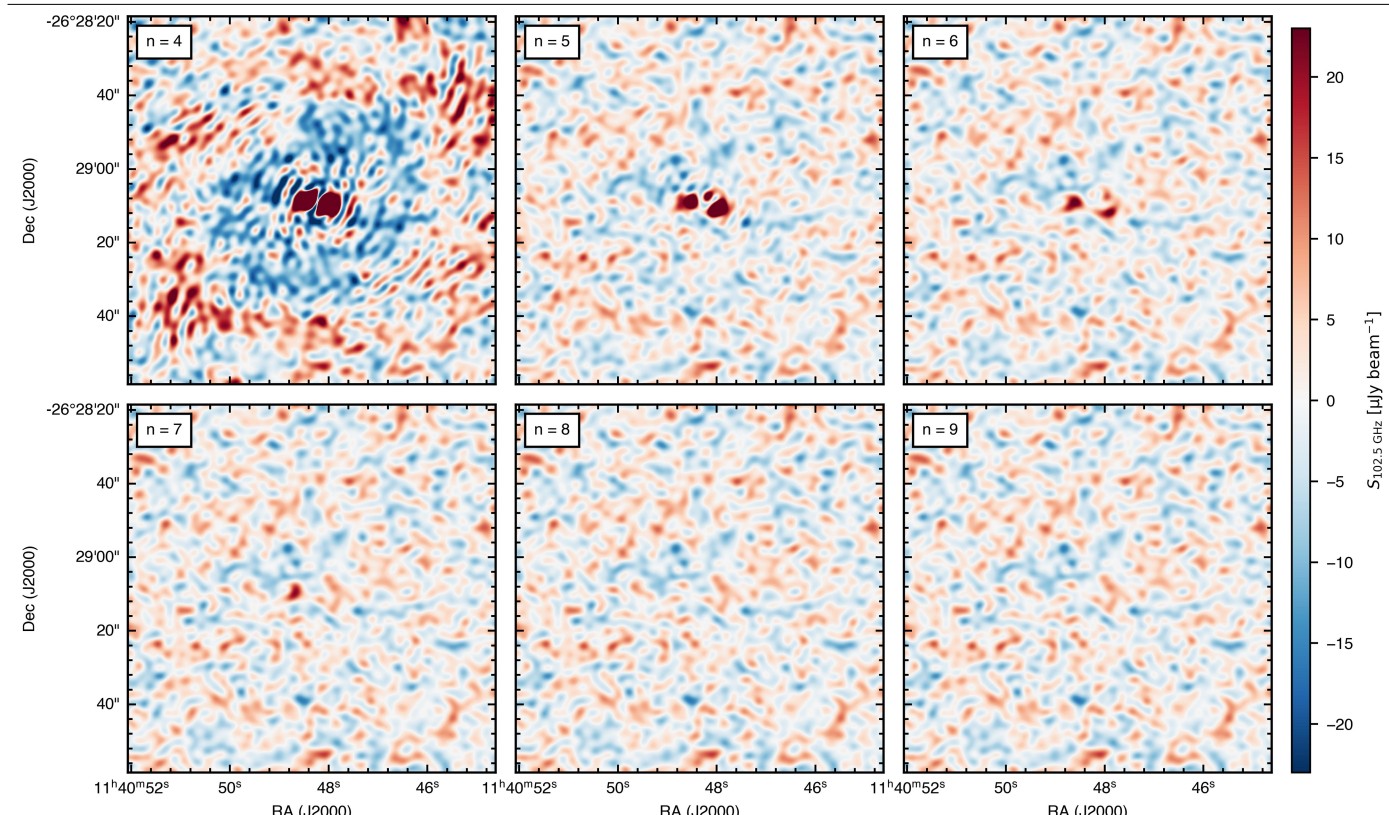

**Extended Data Fig. 4 | Variation of image-space residuals with respect to the number of point-like components used to model the extended radio emission.** Consistent with Extended Data Fig. 1, the images in all panels are generated considering only the data at $uv$ distances smaller than 65 kλ. For the sake of simplicity, we did not apply any deconvolution procedure to the above images. In fact, the large-scale patterns observed in the top-left panel ($n = 4$) reflect the non-uniform response pattern of the available visibility data.

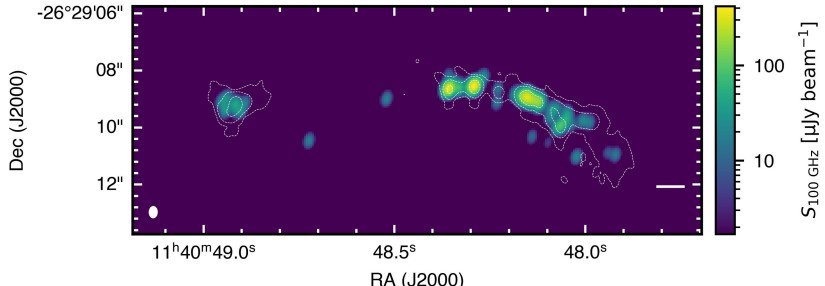

**Extended Data Fig. 5 | Extended radio source from the sparse modelling.**
Shown in the background is the maximum-a-posteriori model of the surface
brightness for the radio source associated with the Spiderweb galaxy obtained
from the sparse-image reconstruction discussed in Methods. The white
contours denote the radio emission as measured from the VLA X-band[8,9] data
(see also Fig. 3). To facilitate the comparison, we smoothed the sparse
ALMA+ACA model to the same angular resolution as that of the VLA data.
Except for a limited number of isolated and low-significance pixels, it is
possible to observe a good agreement between the morphology of the
reconstructed source and the VLA observation.

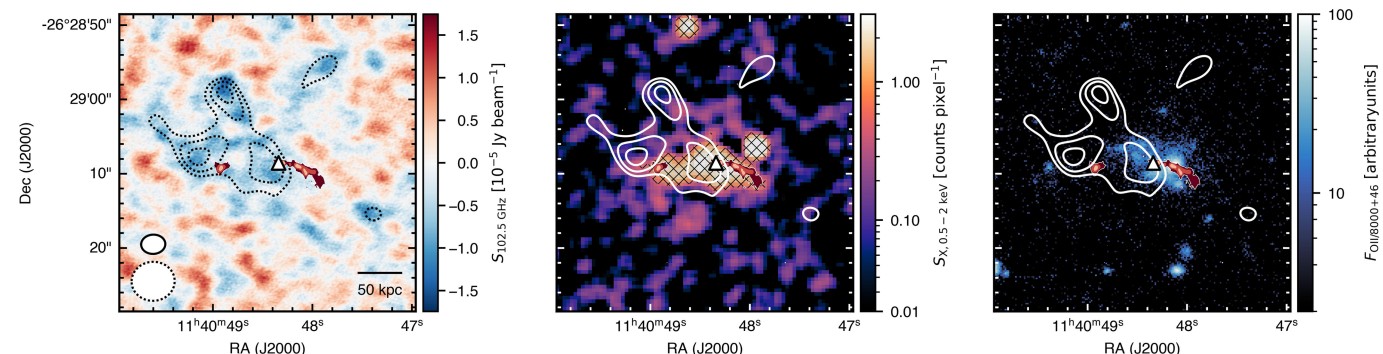

**Extended Data Fig. 6 | Diffuse gas around the Spiderweb galaxy.** The three panels show a comparison between the SZ effect (left) observed in the ALMA+ACA data with the X-ray emission measured by Chandra (centre) and the VLT/FORS1 image[19,30,127] of the extended Lyα nebula (right) within the Spiderweb complex. The ALMA+ACA SZ data are imaged using a natural weighting scheme after subtracting the sparse-modelling solution for the radio source. To reduce the dynamic range of the X-ray map, we suppress the strong emission from the central AGN by fitting and subtracting a point-like model to the data. The hatched region denotes the signal dominated by the inverse Compton X-ray emission associated with AGN or the radio jet[8,9,25], rather than the thermal bremsstrahlung from diffuse hot gas. For reference, we include in dark red the same VLA contours as in Extended Data Fig. 5. When discussing the extended X-ray signal from the ICM, we will then refer only to the region outside this mask (that is, excluding the radio jet). The contours in all the panels trace the decrement in the ALMA+ACA residual image after the application of a 20-kλ Gaussian *uv* taper to enhance the large-scale features associated with the SZ effect.

The levels start at 2σ and increase by a factor of 0.5σ, with σ = 5.30 μJy beam⁻¹, as measured from the corresponding jackknifed image. We mark with a white triangle the position of the Spiderweb galaxy. The ellipses in the lower-left corner of the left panel denote the beams for the ALMA+ACA SZ images at the original (solid ellipse) and *uv*-tapered (dotted ellipse) resolutions. Although the low significance and, in the case of the ALMA data, complex effects associated with interferometric filter hamper the possibility of observing an exact correspondence between the SZ and X-ray images, both of these manifest an asymmetry towards the northeast in their morphological signatures, suggesting an association of the two signals with a hot feature in the diffuse intracluster gas. On the other hand, the Lyα halo occupies the region corresponding to the southwest deficiency observed in both the SZ and the X-ray data. This result might imply that the very core of the Spiderweb galaxy is dominated by cold and warm gas that has not yet been shock-heated to the virial temperature.

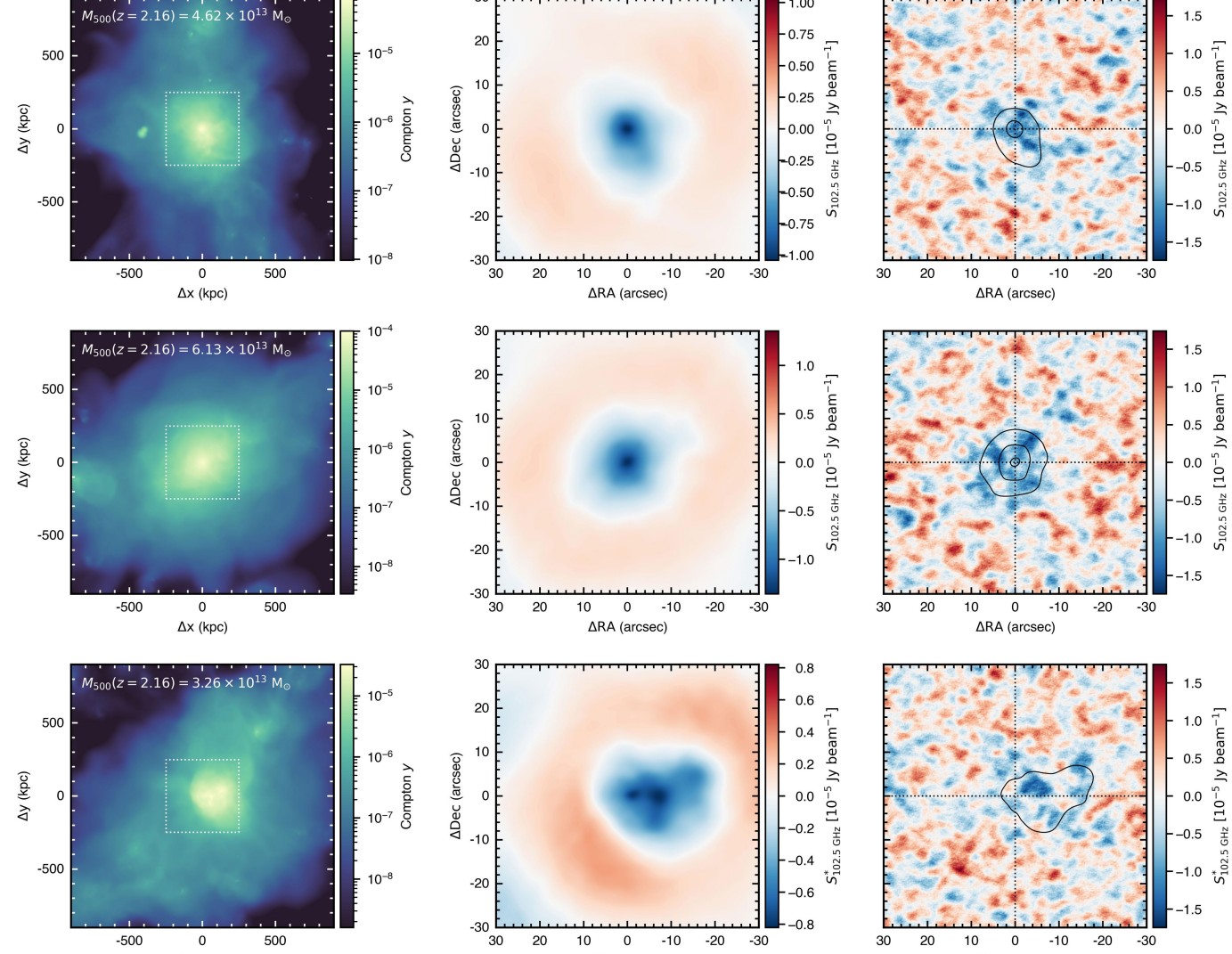

**Extended Data Fig. 7 | Mock observations of simulated galaxy protoclusters.**
Compared are the input Compton $y$ maps from the Dianoga simulation set
(left) and the images reconstructed from the simulated interferometric
observations without (centre) and with (right) adding the jackknife-based
noise realization. All maps are centred on the centroid of the dark-matter halo
of each structure (also denoted as dotted crosshairs in the right panels), roughly
corresponding (modulo a negligible displacement $\lesssim 6$ kpc) to the position of
the most massive galaxy in each system. The top row shows the maps for a
simulated halo selected to generate a SZ signal with amplitude consistent with
that measured for the Spiderweb protocluster. For comparison, we show in the
middle row the most massive halo in the Dianoga sample. The bottom row
corresponds instead to a simulated halo exhibiting a notable offset between
the centre of the dark-matter halo and the peak in the pressure distribution
(that is, SZ signal), with an overall displacement similar to what was found for
the SZ footprint and the central galaxy in the Spiderweb complex. The asterisk
on the colour scale label denotes the case in which the model surface brightness
map is rescaled to match the signal-to-noise ratio of the ALMA+ACA
measurements and, in turn, enhance the corresponding mock observation in
the right panel. The white square in the left panels denotes the field of view of
the mock observation maps in the central and right columns. To facilitate the
comparison, we set the colour map for the right frames equal to the one used in
Extended Data Fig. 6. The solid contours in the right panels trace the noiseless
mock observations in the central column, with contours starting from $1\sigma$ and
increasing by $1\sigma$ steps (for $\sigma = 4.10$ µJy beam$^{-1}$). On the one hand, the limited
signal-to-noise ratio for any of the halos of the sample highlights the need for
deeper observations in view of a proper characterization of the morphological
and physical properties of the forming ICM. On the other hand, the identification
in the Dianoga set of halos with local asymmetries in their pressure distribution
supports the case for the substantial offset observed in the ALMA+ACA data
between the SZ signal and the Spiderweb galaxy not to be representative of
large-scale displacements but the result of a non-trivial combination of
irregular morphology and large-scale interferometric filtering.

**Extended Data Table 1 | Best-fit parameters for the model of the extended source**

| ID | R.A. — | Dec. — | $F_{100\ \mathrm{GHz}}$ [mJy] | Spec. Index — |
|----|--------|--------|-------------------------------|----------------|
| 1 | $11^{\mathrm{h}}40^{\mathrm{m}}46\overset{\mathrm{s}}{.}089 \pm 00\overset{\mathrm{s}}{.}017$ | $-26°29'11\overset{''}{.}026 \pm 00\overset{''}{.}124$ | $0.0331^{+0.0045}_{-0.0045}$ | $3.67^{+0.67}_{-0.81}$ |
| 2 | $11^{\mathrm{h}}40^{\mathrm{m}}46\overset{\mathrm{s}}{.}656 \pm 00\overset{\mathrm{s}}{.}011$ | $-26°29'10\overset{''}{.}787 \pm 00\overset{''}{.}140$ | $0.0208^{+0.0034}_{-0.0034}$ | $3.95^{+0.60}_{-0.64}$ |
| 3 | $11^{\mathrm{h}}40^{\mathrm{m}}48\overset{\mathrm{s}}{.}004 \pm 00\overset{\mathrm{s}}{.}005$ | $-26°29'10\overset{''}{.}542 \pm 00\overset{''}{.}074$ | $0.136^{+0.011}_{-0.014}$ | $-2.60^{+0.27}_{-0.30}$ |
| 4 | $11^{\mathrm{h}}40^{\mathrm{m}}48\overset{\mathrm{s}}{.}112 \pm 00\overset{\mathrm{s}}{.}004$ | $-26°29'09\overset{''}{.}239 \pm 00\overset{''}{.}029$ | $0.842^{+0.027}_{-0.026}$ | $-1.496^{+0.041}_{-0.042}$ |
| 5 | $11^{\mathrm{h}}40^{\mathrm{m}}48\overset{\mathrm{s}}{.}237 \pm 00\overset{\mathrm{s}}{.}008$ | $-26°29'08\overset{''}{.}649 \pm 00\overset{''}{.}020$ | $0.627^{+0.018}_{-0.018}$ | $-1.111^{+0.062}_{-0.061}$ |
| 6 | $11^{\mathrm{h}}40^{\mathrm{m}}48\overset{\mathrm{s}}{.}348 \pm 00\overset{\mathrm{s}}{.}003$ | $-26°29'08\overset{''}{.}632 \pm 00\overset{''}{.}007$ | $0.741^{+0.046}_{-0.035}$ | $-0.635^{+0.041}_{-0.037}$ |
| 7 | $11^{\mathrm{h}}40^{\mathrm{m}}48\overset{\mathrm{s}}{.}676 \pm 00\overset{\mathrm{s}}{.}015$ | $-26°29'09\overset{''}{.}711 \pm 00\overset{''}{.}239$ | $0.0215^{+0.0023}_{-0.0023}$ | $-2.7^{+1.3}_{-1.4}$ |
| 8 | $11^{\mathrm{h}}40^{\mathrm{m}}48\overset{\mathrm{s}}{.}924 \pm 00\overset{\mathrm{s}}{.}002$ | $-26°29'09\overset{''}{.}269 \pm 00\overset{''}{.}017$ | $0.1784^{+0.0023}_{-0.0023}$ | $-2.05^{+0.19}_{-0.20}$ |

We report here the optimal parameters obtained for the model without a SZ term. We note that the inclusion of a SZ component does not introduce any significant variation on the final value of the inferred parameters. A direct comparison between the position of point-like components and the actual morphology of the radio signal in the ALMA+ACA data is provided in Extended Data Fig. 1. As detailed in the text, several point-like components coincide with specific compact sources. In particular, components ID1 and ID2 correspond to distinct protocluster members already reported in the literature[25,34,35,49,62,63]—ERO 284 (ref. [12]) and HAE 229 (refs. [65,66]), respectively. The component ID6 is spatially consistent with the Spiderweb galaxy, whereas ID8 overlaps with the main spot of the eastern radio jet.

**Extended Data Table 2 | Comparison of the best-fit parameters for different SZ models**

| | $\Delta$R.A. ["] | $\Delta$Dec. ["] | $M_{500}$ [$10^{13}$ M$_\odot$] | $r_{500}$ [kpc] | $Y_{\rm sz}(< r_{500})$ $10^{-6}$ Mpc$^2$ | $Y_{\rm sz}(< 5r_{500})$ $10^{-6}$ Mpc$^2$ | $\Delta \log \mathcal{Z}$ — | $\sigma_{\rm eff}$ — |
|---|---|---|---|---|---|---|---|---|
| **A10 UP** | $10.3^{+1.7}_{-1.9}$ | $3.3^{+1.5}_{-1.2}$ | $3.46^{+0.38}_{-0.43}$ | $228.9^{+8.4}_{-9.5}$ | $0.85^{+0.18}_{-0.17}$ | $1.68^{+0.35}_{-0.32}$ | $17.8 \pm 0.5$ | $5.97 \pm 0.08$ |
| **A10 MD** | $10.1^{+1.7}_{-1.8}$ | $3.4^{+1.4}_{-1.3}$ | $3.44^{+0.36}_{-0.42}$ | $228.4^{+8.0}_{-9.3}$ | $0.76^{+0.19}_{-0.17}$ | $1.42^{+0.35}_{-0.32}$ | $19.5 \pm 0.5$ | $6.24 \pm 0.08$ |
| **M14 UP** | $9.9^{+1.7}_{-1.6}$ | $3.7^{+1.3}_{-1.3}$ | $3.73^{+0.48}_{-0.55}$ | $235^{+10}_{-12}$ | $0.78^{+0.19}_{-0.17}$ | $1.57^{+0.38}_{-0.34}$ | $17.9 \pm 0.5$ | $5.98 \pm 0.08$ |
| **M14 NCC** | $10.1^{+1.7}_{-1.7}$ | $3.4^{+1.4}_{-1.3}$ | $3.61^{+0.49}_{-0.56}$ | $232^{+11}_{-12}$ | $0.77^{+0.19}_{-0.17}$ | $1.10^{+0.28}_{-0.25}$ | $18.0 \pm 0.5$ | $6.00 \pm 0.08$ |
| **L15 REF** | $10.8^{+1.5}_{-2.6}$ | $2.4^{+2.1}_{-1.1}$ | $3.26^{+0.34}_{-0.39}$ | $224.4^{+7.8}_{-8.9}$ | $0.70^{+0.14}_{-0.13}$ | $1.57^{+0.31}_{-0.29}$ | $16.0 \pm 0.5$ | $5.66 \pm 0.09$ |
| **L15 8.0** | $10.7^{+1.6}_{-2.0}$ | $2.8^{+1.4}_{-1.2}$ | $5.46^{+0.47}_{-0.55}$ | $266.5^{+7.6}_{-8.9}$ | $1.46^{+0.27}_{-0.27}$ | $4.41^{+0.71}_{-0.71}$ | $21.9 \pm 0.5$ | $6.62 \pm 0.08$ |
| **L15 8.5** | $10.7^{+1.5}_{-2.0}$ | $2.6^{+1.4}_{-1.1}$ | $8.12^{+0.66}_{-0.76}$ | $304.2^{+8.2}_{-9.5}$ | $2.10^{+0.40}_{-0.38}$ | $7.7^{+1.3}_{-1.2}$ | $19.7 \pm 0.5$ | $6.28 \pm 0.08$ |
| **G17 EXT** | $9.9^{+1.7}_{-1.8}$ | $3.7^{+1.4}_{-1.3}$ | $3.08^{+0.36}_{-0.41}$ | $220.2^{+8.6}_{-9.8}$ | $0.70^{+0.15}_{-0.14}$ | $1.15^{+0.23}_{-0.22}$ | $19.6 \pm 0.5$ | $6.26 \pm 0.08$ |

We refer to the nested sampling section in Methods for specific details of the models adopted in our analysis. The coordinates of the SZ components are reported as shifts with respect to the position of the Spiderweb galaxy (11h 40m 48.34s, −26° 29′ 08.55″). The radii $r_{500}$ are computed from the reported $M_{500}$ values as $r_{500} = \sqrt[3]{M_{500}/\left[\frac{4\pi}{3}500\rho_{\rm c}(z)\right]}$, in which $\rho_{\rm c}$ is the critical density of the Universe at redshift $z$. The log-evidence difference $\Delta\log\mathcal{Z}$ for each of the listed models is computed with the respect to the case comprising only the extended radio source. The corresponding estimates for the effective significance $\sigma_{\rm eff}$ are computed under the assumption of a multivariate normal posterior distribution (that is, $\sigma_{\rm eff} = {\rm sgn}(\Delta\log Z)\cdot\sqrt{2\,|\Delta\log Z|}$).