## [Peer Review File · Nature]

Manuscript Title: Forming intracluster gas in a galaxy protocluster at a redshift of 2.16

Reviewer Comments & Author Rebuttals

Reviewer Reports on the Initial Version:

Referee #1 (Remarks to the Author):

The manuscript, "Direct detection of hot intracluster gas within an archetypal protocluster of galaxies at $z \approx 2.16$ " by Di Mascolo et al. represents new results purporting to detect the S-Z signal from the well-studied radio galaxy and over density, MRC 1138-262. Such a detection is extremely important as it would indicate a hot ionized medium early in the formation of a proto-cluster. The observations, reduction, and analysis are challenging, not only due to the faintness of the signal, but also because the targeted region is centered on a radio galaxy bright both in the radio and submm/mm. I note particularly the care in which the authors take in both reducing and analysing these data.

The results are challenging to understand. It is a rather technical paper for Nature, which is not disqualifying, but does make it difficult to provide a clear message as to why this result is significant to a general audience. I view this as the most significant weakness of the manuscript.

My main concern in the paper is the fact that, while it may be significant statistically, the fitting is done assuming that the decrement regions are largely symmetrical and close to equilibrium. This is evident in Fig. 8 of the methods section. It appears as if none of the simulations as similar the data in the image plane and the ones shown are fitted by a single symmetric source. Of course, in the binned UV profiles, as the authors note, the fact that the centers are offset from the pointing center and has multiple components are not important for the fitting of the amplitudes. It is in the imaging plane where these features become evidence. I am not suggesting this isn't real structure but the interpretation of it of course depends on both the model assumptions and what is in the simulation. The analysis, if warranted by the data, will naturally break up into multiple roughly symmetric components. I think the fact that the exact structure seen does depend on the model and thus should be presented more forthrightly.

It is for this reason that I believe the presentation of the results at times reads rather contradictory. On the one hand, some paragraphs indicate that the gas is in approximate equilibrium and a good or fair probe of the dark matter halo (substructures) but in others, you must argue that it is not in an overall equilibrium because there is structure. I think outlining better what we expect from the early evolution of clusters in the first paragraphs (complex structure, accretion of gas, possible accretion of sub-structure, etc), and then say that in order to interpret these challenging data, that you rely on some simple assumptions which may imprint rather symmetric features in the image. And despite saying the results are qualitative, you do quote masses of the dark matter halos.

While it is not clear to me if it is reasonable to expect the radio galaxy to be the center of the potential, the authors clearly did. In fact, in Carilli et al. 2022, it was argued that the radio jets are

ram pressure confined. At an advance speed of-order a few 1000 km/s, it is logical to think that there is a hot halo surrounding the radio galaxy. Even if the confining material was neutral or warm ionized gas (the Ly-alpha halo), the passage of a jet will cause some of the gas to reach temperatures sufficiently high to result in an S-Z decrement. Instead, the S-Z decrement is offset significantly from the radio galaxies and on the opposite side of where the bent jet lies. Doesn't this imply that heating by the jet is not significant? I note that the authors discuss the complex interaction of the radio jets with its surroundings and yet find no concrete evidence for this. Other explanations for why it may be happening but is not observed using the S-Z effect are of course possible, and the authors generally mention them. But again, without some sort of explanation, I have some doubts about the nature of the signal seen in the data or I suppose more specifically, the authors interpretation of the signal.

Referee #2 (Remarks to the Author):

Dear editor, dear authors,

The manuscript "Direct detection of hot intracluster gas within an archetypal protocluster of galaxies at $z = 2.16$ " claims the first detection of intracluster gas in the direction of a protocluster located in the Spiderweb complex, at $z=2.156$. The thermal Sunyaev-Zel'dovich effect (tSZ), induced by the thermal pressure of the intracluster electrons, is detected as the missing signal explaining the discrepancy between the data and a model including only the radio source model. The manuscript presents a convincing detection: Different tSZ models are considered to assess robustness, and only the most conservative result is cited. As a direct detection of the virialized(ing) gas in a protocluster, this result is significant for the astrophysics community as a whole. Additionally, the fact that the halo mass and radius obtained by the authors with the SZ observation are different from previous measurements shows the potential for such observations to tell us more about the formation of galaxy clusters.

However, the physical interpretation the authors make of this detection is ambiguous, and no clear conclusion about it can be found in the manuscript – the final sentence of the abstract being rather confusing for the reader. The authors spend some time in the "Methods" section looking for explanations for their low halo mass estimate, for example investigating the possibility of signal coming not from one main halo but several sub-haloes, but barely refer to this in the main text. Notably, the authors claim the "statistical insignificance" of the shift between the SZ models and the data points in fig 1b, but I am left unconvinced of the fact that it is not a modelling problem. The authors should refine this analysis to reach a more convincing conclusion and mention both these points (the signal shift and the mass discrepancy) in the abstract.

Additionally, the statistical estimators cited throughout the manuscript (error bars, Bayesian evidence, detection significance) are overall poorly defined and difficult to trace (e.g., a number cited in the main text does not appear in the table referred to in the same paragraph). I would appreciate if the authors were more explicit about the tools they use and the choices they make – such as the "best" and "worst" models in fig 1b.

Once the authors have addressed these two points, and the related comments that I detail below, I will be happy to consider this manuscript for publication again.

Note that, being not familiar with proto-clusters observations, I assessed the methods used in this manuscript, as well as the validity of the results presented, but I am not in a position to assess their originality, the observational strategy employed (e.g., calibration) or the use of appropriate references by the authors.

Kind regards.

Extraction of the SZ signal with nested posterior sampling

The use of different SZ models to explain the difference between the measured data points and the model including the radio source alone (fig 1b) builds confidence in the detection. However, there are some issues the authors should address:

1. The errors on the radio source model shown in figure 1a (defined as “the standard deviation of the scatter within each uv bin due to combination of the elongated morphology of the radio galaxy and the asymmetry in the visibility patterns of the data” which is not clear to me), are never mentioned again in the analysis and do not seem to be propagated – if they are, the authors should make it clearer in the text. Indeed, since the errors are about 0.1 mJy large, taking them into account can lower the significance of the SZ detection.
2. Different numbers of compact components are used to model the radio source and, from my understanding, the “best” of these models is chosen before including the SZ contribution in the model (lines 82-92, page 5). If so, could the authors please make sure the results you cite about the SZ signal are marginalised over the number of compact components, or at least indicate how much the modelled SZ signal changes when source components are added or removed? Could they also confirm that the “model uncertainties” cited in the caption of figure 1a correspond to the variance between models including various numbers of compact components?
3. Overall, a more complete description of the source model fit would be appreciated.
4. The effective significance cited in the main text (line 4, page 2), does not correspond to any of the values listed in Table 1 for the different SZ models considered.
5. The authors do not specify why the “best” and “worst” SZ models cited in the caption of figure 1b are considered as such.
6. Could the authors explicit how “broadly consistent with unity” the calibration scaling parameters are (within the 5% imposed? – line 35, page 6).

Theoretical interpretation and comparison to previous mass studies

Despite representing a large fraction of the “Methods” section, the discussion of the SZ-measured halo mass compared to previous measurements is barely mentioned in the main text. Although I understand no clear conclusion can be drawn, this analysis could probably be refined, and/or tentative explanations could be made clearer in the text (namely in the abstract).

1. Could the authors add a column to table 1 presenting the Y_{SZ} values for each model to support the discussion of page 7?
2. In fig 2, the models with larger halo mass lead to stronger negative SZ signal (and supposedly a Y_{SZ} value closer to the one obtained from velocity measurements?). In figure 1b, you present SZ models that underfit the data points and lead to a lower mass than expected from other

observations. Could the authors comment on this?

3. The way the authors investigate the potential for several sub-haloes to contribute to the observed signal rather than only one halo is extremely interesting. Would it be possible to include a slightly more physical model of sub-halo mass distribution rather than subcomponents of equal mass (line 34, page 7) – maybe something as simple as a power law?

4. Would it be possible to allow for more than one SZ (pressure) component in the Bayesian analysis? The strength of the argument made on line 53 of page 7 about the posterior velocity distribution not being bimodal is dubious to me if several components are not allowed in the analysis in the first place.

Typos & general comments

1. Please give the definition of the log-evidence used in this work.

2. Could the authors explain in the text their conclusion about the virialization of the gas? It does not seem to me that the statement “the identification of virialized subhaloes through a spectroscopic characterization is ineffective” made by the authors (line 25, page 3) is properly backed.

3. A flat prior can have strong effects on your posterior distributions and be far from “uninformative” (line 91, page 5).

4. Typos: lines 7 and 8, page 3; line 81 page 11

5. The abstract is clear, well-written, and showcases the importance of this detection well. However, the last sentence, related to the theoretical interpretation, is vague, unconvincing, and difficult to relate to the analysis presented in the main text.

6. It does not seem that the availability of the ACA data used in this work is listed.

7. Could the authors explain how the protocluster considered in this work is archetypal?

Author Rebuttals to Initial Comments:

In the report below, the comments from Referee 1 are reported in red, our responses in black.

REFEREE 1

The manuscript, “Direct detection of hot intracluster gas within an archetypal protocluster of galaxies at $z \simeq 2.16$ ” by Di Mascolo et al. represents new results purporting to detect the S-Z signal from the well-studied radio galaxy and over density, MRC 1138-262. Such a detection is extremely important as it would indicate a hot ionized medium early in the formation of a proto-cluster. The observations, reduction, and analysis are challenging, not only due to the faintness of the signal, but also because the targeted region is centered on a radio galaxy bright both in the radio and submm/mm. I note particularly the care in which the authors take in both reducing and analysing these data.

We thank the referee for the very positive response. We agree, the detection of a hot (proto-)ICM is indeed the crucial implication of the SZ observations here. To better emphasize this message, we adapted the conclusive statements in the main text.

The results are challenging to understand. It is a rather technical paper for Nature, which is not disqualifying, but does make it difficult to provide a clear message as to why this result is significant to a general audience. I view this as the most significant weakness of the manuscript.

We have attempted to restructure the work to emphasize the salient results clearly, while relegating the most technical aspects to the appendix. In particular, as mentioned above, we performed a major refactoring of our conclusions, trying to emphasize the main result of this work (i.e., the detection of the SZ signal from the Spiderweb galaxy and, thus, the direct confirmation of theoretical predictions and indirect observational evidences for the presence of a forming ICM).

My main concern in the paper is the fact that, while it may be significant statistically, the fitting is done assuming that the decrement regions are largely symmetrical and close to equilibrium. This is evident in Fig. 8 of the methods section. It appears as if none of the simulations as similar the data in the image plane and the ones shown are fitted by a single symmetric source. Of course, in the binned UV profiles, as the authors note, the fact that the centers are offset from the pointing center and has multiple components are not important for the fitting of the amplitudes. It is in the imaging plane where these features become evidence. I am not suggesting this isn't real structure but the interpretation of it of course depends on both the model assumptions and what is in the simulation.

The necessity of employing spherically symmetric models is indeed a major limitation and resulting from the need of dealing with the modest sensitivity of the available data. As discussed below, we however report the results obtained when including multiple haloes or by relaxing the spherical assumption allowing for ellipsoidal distributions. From a statistical point of view, any potential mismatch between the model and the actual SZ distribution would result in a reduction of the statistical significance. As such, the reported significance represents only a lower limit to the actual value, thus strengthening the reported detection. To be as conservative as possible, we further consider the model with the lowest significance (A10 UP; $\sigma_{\text{eff}} = 5.97 \pm 0.08$) as a reference (page 2, lines 39-43).

Regarding the comparison with simulations, we would like to emphasize a few key points.

- The simulations are never employed as reference for the model reconstruction from the ALMA+ACA data, but are used only for gaining a theoretical sense about the amplitude of the SZ signal from any (sub)structures associated to protocluster systems (here defined as physical complexes that are known to evolve into massive clusters at $z = 0$). As already noted in the text, the goal is thus not to find a simulated protocluster exactly matching the morphological properties of the Spiderweb protocluster.
- We are never fitting the simulations, but only projecting the simulated SZ maps onto the ALMA+ACA uv measurements to perform the comparison discussed above. In fact, the middle and right panels of Extended Data Fig. 7 (previously Fig. 8) are showing mock observations of the simulated from the left

Figure R1.1: Comparison of the Compton y maps from the “Dianoga” simulated halos. Each set of three panels comprise the projection of the same halo along three different projection directions. It is possible to immediately see how a large fraction of structures exhibits disturbed morphologies, deviating from spherical symmetry.

Figure R1.2: Zoomed-in version of Fig. R1.1, focusing on the central 500 kpc to highlight the morphological asymmetries on small scales. The select region is consistent with the field of view employed in, e.g., Figs. 4 and 9 in the manuscript.

panels, to which we are applying the transfer function of the available radio-interferometric data without performing any modelling step. Any ICM features, if on scales that are probed by the ALMA and ACA data, are thus reflected in the mock images.

- In Extended Data Fig. 7 (previously Fig. 8), we showed only two (now three) out of a total of 27 distinct haloes analyzed for performing the comparison in Fig. 2. The selection of these two specific structures was arbitrary, and motivated only by the desire to show, in one case, a protocluster structure with an overall SZ amplitude (and, thus, mass, not morphology) similar to the one observed for the Spiderweb protocluster and, in the other case, the most massive halo in the sample (we emphasise this information at page 15, lines 65-78). To provide an additional, representative example, we added a third row showing a halo exhibiting a displacement between the SZ peak and the central galaxy similar to what observed in the Spiderweb protocluster. A glimpse at the SZ morphologies for all the haloes considered for our analysis is provided in the mosaic of Fig. R1.1. In particular, it is possible to note that many structures exhibit fairly irregular morphologies, characteristic of disturbed/merging states.
- The excessive regularity in the SZ maps obtained from the simulations is only emphasised by the specific choice for the colour scale. This specifically highlights the SZ footprint on large scales, where the dominant contribution in shaping and driving the assembly of the proto-ICM is provided by the gravity from the main (hence relatively symmetric) dark matter potential. Despite of this, asymmetric features are visible also on protocluster scales ($\gtrsim 200$ kpc; see the left panels in Extended Data Fig. 7, previously Fig. 8, and mosaic in Fig. R1.2).

As a final remark, we stress that all the radio-interferometric images employed in our study are provided mostly for visualisation purposes. In fact, the significant correlation in the image-space noise, and the complex transfer function resulting from the sparse coverage of the visibility plane (mostly impacting diffuse signals as the SZ effect) do not allow for a straightforward and reliable interpretation of any image-space features independently of more accurate analyses.

The analysis, if warranted by the data, will naturally break up into multiple roughly symmetric components. I think the fact that the exact structure seen does depend on the model and thus should be presented more forthrightly.

To address this type of concern in particular, we tried adding additional SZ terms with the same prior distributions as the single-component case, but adding alternatively an ordering constrain on the right ascension or the declination directions (to avoid any label switching issue). Below is a brief summary of our findings.

- In the case $n = 2$, the sampler converges to a solution that adds an off-center component to the main SZ feature reported in our manuscript (Fig. R1.3). Specifically, this falls around $\sim 27.3''$ (226 kpc) south-east of the Spiderweb galaxy. We observe that this structure is independent of the specific choice for the coordinate parameter on which to impose the ordering assumption. As discussed in the text, the distance from main SZ halo is of the order of the r_{500} of the latter, implying that the secondary component corresponds to a halo that is marginally detached from the main SZ structure. However, the limited significance of this secondary SZ feature ($\sigma_{\text{eff}} \lesssim 1.90$) and the lack of correspondence with any galaxy overdensity or X-ray signal limit the possibility of a firm assessment of its origin.
- On the other hand, increasing the number of SZ components to $n = 3$ results in a degradation of the significance of the model.

Similarly, allowing for an ellipsoidal pressure distribution provides unphysical results, with an ICM distribution having a plane-of-sky eccentricity of ~ 0.96 (i.e., a minor axis being only 4% of the major axis).

All the results mentioned above mainly imply that the available data do not have enough constraining power for providing statistically meaningful constraints on the morphological properties of the forming ICM. In this sense, the reported detection thus represents a conservative estimate. The marginal identification of a secondary SZ peak, however, does not undermine our results and, in fact, strengthen the significance of our SZ detection. The moderate displacement with respect to the main SZ halo in fact implies that any deviation from spherical symmetry can not be meaningfully described by the addition of

a second component (as in the case of the extended radio source), while supporting all the evidences for the disturbed nature of the proto-ICM. We added in Methods a discussion on the above multi-halo modelling and respective results (“Multiple SZ components”; pages 9-10).

Figure R1.3: Results from the modelling runs including two SZ components. In this figure is reported the result specifically obtained when considering our “best” model ($L15 = 8.0$). The red and black (almost identical) contours denote the marginalized posteriors resulting from the modelling runs with an ordering prior on the right ascension and declination parameters, respectively. For reference, we report in blue the posterior for the corresponding best-fit one-component model reported in the manuscript.

It is for this reason that I believe the presentation of the results at times reads rather contradictory. On the one hand, some paragraphs indicate that the gas is in approximate equilibrium and a good or fair probe of the dark matter halo (substructures) but in others, you must argue that it is not in an overall equilibrium because there is structure. I think outlining better what we expect from the early evolution of clusters in the first paragraphs (complex structure, accretion of gas, possible accretion of sub-structure, etc), and then say that in order to interpret these challenging data, that you rely on some simple assumptions which may imprint rather symmetric features in the image. And despite saying the results are qualitative, you do quote masses of the dark matter halos.

We thank the referee for the suggestion. We integrated the corresponding text at the beginning of our manuscript (page 1, lines 22-32; page 2, lines 20-21).

We would like to stress, anyway, that the deviation from equilibrium expected in disturbed systems like a galaxy protocluster is factored in all our discussions. For instance, the mismatch between the SZ-based mass estimate and the value from the literature obtained from velocity dispersion measurements is used not to highlight an issue with the latter, but as a modelling proxy of the extremely dynamic state of the system (and, thus, of the impossibility of correlating the SZ signal to a reliable mass estimate). In fact, all the mass estimates are nevertheless employed mostly as a quantitative, despite approximate indi-

Figure R1.4: Comparison of the high-resolution images obtained after subtracting only the radio source model obtained without (left) and when adding (right) an SZ component. As such, the underlying SZ contribution is still present in both panels. The difference between the two images is barely visible, suggesting a minor contribution by the SZ model in driving the reconstruction and, thus, in imprinting any specific pattern in the final images.

cation of the amplitude of the SZ signal. We now better clarify this point, e.g., at page 2, lines 1-15. Further, a thorough discussion of the shortcomings of the hydrostatic equilibrium assumption is provided in Methods (“Systematics in the Bayesian analysis”; page 11). Nevertheless, we note that, despite the clear limitations of our models (which we have attempted to make explicit throughout the manuscript), the rough agreement between the mass derived with our parametric modelling and the one of the simulated halos with SZ signal analogous to what is measured in the ALMA+ACA data provides additional confidence in our reconstruction. This is the only part of our discussion that presents qualitative arguments, although not on the halo masses but just regarding the specific ICM features we could expect in $z \sim 2$ protoclusters (we add a statement to specifically emphasize this; page 3, lines 23-25).

In regards to the possibility of imprinting symmetric features in the images, we note that, as already mentioned above, we do not use radio-interferometric images for any our quantitative considerations (especially in the case of the low-resolution parametric analysis), but provide them solely as a visual aid to facilitate discussion. We further note that the consistency between the radio source models reconstructed with and without including an SZ component implies that any assumption on the SZ distribution has little (if any) impact on the radio source model itself. In turn, no symmetric features are expected to be imprinted in the radio model or in the maps resulting from the subtraction of the radio model. In fact, when considering the non-parametric high-resolution reconstruction, the SZ footprint of the Spiderweb protoclusters is observed to be strongly asymmetric, independently of the spherically symmetric assumption for the SZ model. Also, the same SZ pattern is observed also when generating a radio source-subtracted image from the model not comprising an SZ component, modulo marginal differences right at the position of the radio source (see comparison in Fig. R1.4). Finally, we stress once more that any underlying asymmetric feature which is not included in our models will only act in the direction of increasing the significance of the detection.

While it is not clear to me if it is reasonable to expect the radio galaxy to be the center of the potential, the authors clearly did.

All the results reported in our manuscript do not rely on the assumption for the radio galaxy to be at the center of the gravitational potential, and, as such, they are valid independently of any considerations regarding of the spatial displacement between the SZ effect and the Spiderweb galaxy (see the main text and the discussion in “Nested posterior sampling” and “Sparse imaging” sections for details on the modelling assumptions). In fact, such an offset is now mentioned only once in the main text (page 3, lines 6-7), and in relation to the evidence for the disturbed state of the proto-ICM.

We further note that the case for the radio galaxy spatially coinciding with the centre of the gravitational

potential is mentioned in a broader context discussing the multi-wavelength constraints available on the Spiderweb complex. At line 101 of page 13, we state that such an assumption is valid only in the case of relaxed systems, as observed, e.g., in cool-core clusters at redshifts lower than the Spiderweb protocluster. Clearly, as broadly discussed in section “Insights from multi-wavelength information” (pages 13-14), this does not apply to the Spiderweb complex. In this regards, the observation of the displacement is supporting and complements the picture of the proto-ICM in the Spiderweb complex to be experiencing energetic and dynamically complex processes (see the corresponding, aforementioned section). To better emphasise that we *do not* expect the Spiderweb galaxy and the centroid of the gravitational potential to correspond spatially, we added an explanatory statement at line 20-24 of page 13.

In fact, in Carilli et al. 2022, it was argued that the radio jets are ram pressure confined. At an advance speed of-order a few 1000 km/s, it is logical to think that there is a hot halo surrounding the radio galaxy. Even if the confining material was neutral or warm ionized gas (the Ly-alpha halo), the passage of a jet will cause some of the gas to reach temperatures sufficiently high to result in an S-Z decrement. Instead, the S-Z decrement is offset significantly from the radio galaxies and on the opposite side of where the bent jet lies. Doesn't this imply that heating by the jet is not significant? I note that the authors discuss the complex interaction of the radio jets with its surroundings and yet find no concrete evidence for this. Other explanations for why it may be happening but is not observed using the S-Z effect are of course possible, and the authors generally mention them. But again, without some sort of explanation, I have some doubts about the nature of the signal seen in the data or I suppose more specifically, the authors interpretation of the signal.

As correctly suggested by the referee, the lack of a spatial correspondence between the SZ effect and the radio galaxy is potentially consistent with a subdominant jet contribution to the overall ICM heating. In fact, in our manuscript we *do not* explicitly advocate for the AGN origin for the mechanism heating the proto-ICM environment. Our work is instead entirely devoted to solely report the detection of the SZ effect in the direction of the Spiderweb protocluster, which is providing the observational confirmation for the presence of a forming halo of thermalising ICM. Although we agree with the referee on the importance of a fully energetic analysis, constraining if and at what level AGN feedback or gravitational process contribute to increasing the thermal energy of the system is unfortunately not feasible with current data and out of the scope of this paper. Gaining a clear understanding of all the specific contributions driving the ICM heating process and defining the overall thermal energy budget of the system would require a dramatic improvement in the available observations (for the specific SZ case, in terms of sensitivity and frequency coverage). For the same reason, it is not possible to use the available SZ data to provide a conclusive answer on the mechanism governing the interaction between the forming ICM and the energetic AGN in the Spiderweb protocluster. However, we would like to mention a few important points.

- The reported discussion on the interactions between the ICM and the radio jet are presented with the purpose of including the SZ information in the complex picture built through the years in the extensive literature on the Spiderweb galaxy and its relation with the environment. All the evidences supporting such a connection are thus derived from previous works, and exploited to link our finding (i.e., the presence of a hot ICM halo in the Spiderweb protocluster) to the broader context of the evolution of protoclusters. Most importantly, the SZ detection is not reported to provide concrete evidence for such connection, but, on the contrary, is derived independently of it and interpreted a posteriori in a multi-wavelength perspective.
- The direct comparison of the SZ signal with the radio galaxy clearly shows that the characteristic scales of the two signals differ by almost an order of magnitude — i.e., with the SZ halo extending over scales of the order of $r_{500} \gtrsim 200$, and the radio lobes being at ~ 60 kpc from the central galaxy and having a transverse width $\lesssim 20$ kpc. An enhancement of the SZ signal associated to the mechanical action of the radio jet on the proto-ICM will thus be mostly confined within such scales, and expected to be roughly cospatial with the extended radio galaxy. We are not able to find evidence (e.g., signatures of a cocoon

Figure R1.5: Comparison of the input SZ map (left) for simulated cluster from the Dianoga simulation set, and the corresponding image (right) resulting from the application of ALMA+ACA filtering.

shock around the Spiderweb galaxy) that would support a direct impact of the AGN on the ICM heating, but the sensitivity and resolution of the available ALMA+ACA data can not exclude such a possibility as a whole. In order to gain a quantitative (although approximate) sense of the AGN contribution to ICM heating, we perform simple comparison of the expected energy budget for the ICM and AGN using numerical estimates from Carilli et al. 2022 and Tozzi et al. 2022b. From the latter, we can calculate the thermal energy within ~ 60 kpc from the Spiderweb galaxy (i.e., the rough extent of the radio jets) as $E_{\text{th}} = \frac{3}{2}P_{\text{th}}V$, where V is assumed equal to the spherical volume $\frac{4\pi}{3}r^3$ of radius $r = 60$ kpc around the radio galaxy. Assuming for ions and electrons to have same temperature, the total thermal pressure P_{th} is instead simply derived as $P_{\text{th}} = P_{\text{i,th}} + P_{\text{e,th}} \approx 2n_eT_e$ from the density and temperature estimates for the central ICM region by Tozzi et al. 2022b, i.e., $n_e = 1.4 \times 10^{-2} \text{ cm}^{-3}$ and $T_e = 2.4 \text{ keV}$. This provides $P_{\text{th}} \simeq 9 \times 10^{-11} \text{ erg cm}^{-3}$, in turn resulting in $E_{\text{th}} \simeq 3 \times 10^{60} \text{ erg}$. As for the jet, we refer to the non-thermal pressure estimate by Carilli et al. 2022 ($P_{\text{nth}} \simeq 9 \times 10^{-10} \text{ erg cm}^{-3}$) a non-thermal and assume the jet volume to be described by cylinders 25 kpc and 60 kpc in width and height, respectively (based on the VLA morphology of the jet structure; Carilli et al. 2022). The corresponding energy is then $E_{\text{nth}} = 3P_{\text{nth}}V \simeq 6 \times 10^{60} \text{ erg}$, of the same order of magnitude of the thermal energy estimate. This implies that, assuming that half of the jet energy goes into the shocked thermal gas and half stays in a non-thermal cocoon (e.g., Gaibler et al. 2009), the total SZ signal within ~ 60 kpc around the Spiderweb galaxy can be boosted by less than a factor of 2. Although these remain approximate estimates, such a comparison directly supports the conclusion that any local enhancement would not necessarily be observed in the current data. As we think that the point above might be important, we added a short statement in Methods (lines 40-52, page 14).

We finally note that the orientation of the ellipsoidal SZ model (see above) does not correlate with the direction of the radio galaxy, excluding both that the elongated model is actually tracing AGN-heated gas and that the fitting is driven by the subtraction of the radio source.

- Multi-wavelength observations have been providing evidence for the Spiderweb radio galaxy to be transitioning between the quasar- and radio-mode regimes (we refer to Tozzi et al. 2022b for a summary). This implies that the efficiency of mechanical feedback might be not as efficient as observed, e.g., in local clusters.
- In relation to the pressure confinement of the radio jet hypothesised in Carilli et al. 2022, we note that we actually report in section “Insights from multi-wavelength information” in Methods (page 14, lines 27-49) a complementary scenario, in which a dynamical action imprinted by the motion of the Spiderweb galaxy within the extended ICM halo might contribute to enhancing the jet asymmetry.

All in all, despite multiple evidences are supporting the subdominant role of AGN feedback in providing the energy source required to heat the forming ICM, we can not perform any robust study of its actual efficiency. As such, we prefer to include just a mention of such a possibility (page 14, lines 33-59), without attempting any focused, but potentially inaccurate analysis.

As a final remark, we note that the displacement between the ICM and the Spiderweb galaxy might be accentuated by radio-interferometric filtering. We provide in Fig. R1.5 a comparison between the input and filtered SZ maps for a Dianoga cluster. The bulk of the SZ signal is suppressed as a consequence of

the limited large-scale sensitivity of ALMA+ACA, while the small scale, peaked features are preserved. The measured signal will thus appear as small "SZ islands", i.e., less extended than one would expect when considering the whole ICM distribution.

In the report below, the comments from Referee 2 are reported in red, our responses in black.

REFEREE 2

Dear editor, dear authors,

The manuscript “Direct detection of hot intracluster gas within an archetypal protocluster of galaxies at $z = 2.16$ ” claims the first detection of intracluster gas in the direction of a protocluster located in the Spider-web complex, at $z=2.156$. The thermal Sunyaev-Zel’dovich effect (tSZ), induced by the thermal pressure of the intracluster electrons, is detected as the missing signal explaining the discrepancy between the data and a model including only the radio source model. The manuscript presents a convincing detection: Different tSZ models are considered to assess robustness, and only the most conservative result is cited. As a direct detection of the virialized(ing) gas in a protocluster, this result is significant for the astrophysics community as a whole. Additionally, the fact that the halo mass and radius obtained by the authors with the SZ observation are different from previous measurements shows the potential for such observations to tell us more about the formation of galaxy clusters. However, the physical interpretation the authors make of this detection is ambiguous, and no clear conclusion about it can be found in the manuscript – the final sentence of the abstract being rather confusing for the reader.

We thank the referee for the positive response. We agree that the past version of the abstract was presenting an ambiguous conclusion. We adapted it to reflect the main discussions presented in the main text. In particular, we included mentions both to the mass estimates and their discrepancy with previous studies, and to multi-wavelength evidences for the highly disturbed state of the system. Regarding the former point, we note that we preferred to set the discussion in terms of SZ amplitude rather than mass, being the latter a matter of extrapolation from mass-observable relations and given the impossibility of clearly stating in abstract all the caveats necessary to correctly interpret the mass estimate.

The authors spend some time in the “Methods” section looking for explanations for their low halo mass estimate, for example investigating the possibility of signal coming not from one main halo but several sub-haloes, but barely refer to this in the main text.

We extended the main text according to referee’s suggestions, incorporating more details on the multi-halo considerations (starting from line 59 at page 2). For the sake of readability, we nevertheless decided to keep reporting all the technical details in the corresponding Methods subsections.

Notably, the authors claim the “statistical insignificance” of the shift between the SZ models and the data points in fig 1b, but I am left unconvinced of the fact that it is not a modelling problem. The authors should refine this analysis to reach a more convincing conclusion and mention both these points (the signal shift and the mass discrepancy) in the abstract.

As more broadly discussed below, the observed shift was in a major part caused by a bug in the plotting routine used for producing Fig. 1b, now updated to the correct form. We nevertheless refer to rest of this report for the discussion of various tests we performed to assess for any modelling issue.

Regarding the mass discrepancy, as noted in the previous comment, we included a mention to it in the abstract, refactoring the argument in terms of amplitude of the SZ signal.

Additionally, the statistical estimators cited throughout the manuscript (error bars, Bayesian evidence, detection significance) are overall poorly defined and difficult to trace (e.g., a number cited in the main text does not appear in the table referred to in the same paragraph). I would appreciate if the authors were more explicit about the tools they use and the choices they make – such as the “best” and “worst” models in fig 1b.

At page 2 (lines 30-39), we define all the statistical quantities relevant for the discussion presented in the manuscript (i.e., “bayesian evidence” and “log-evidence”, “effective significance”). We further corrected

Figure R2.1: Comparison of the binned uv data and corresponding models for different directions in the uv plane (adapted from the left panel of Fig. 1 in the main text). The green regions in top-right insets show the specific quadrants of the uv plane that were considered for computing the binned profiles.

and generalised the statement at lines 23-25 to clarify that the 16th-50th-84th percentiles are used for defining best-fit values and corresponding uncertainties for all the model parameters, not only the halo mass M_{500} .

Regarding the definition of “best” and “worst” models, these refer to the pressure models that are most and least favoured from a statistical point of view (i.e., considering their Bayesian evidence; Tab. 2). We modified the corresponding reference in the caption of Fig. 1b to clarify this.

Finally, we agree with the referee that we did not successfully describe the tools we used in our analysis. In particular, we realised that we did not provide enough information regarding the methods actually employed to perform our parametric analysis. We thus modified the introduction of the “Nested posterior sampling” subsection and provide a brief summary of the tool (lines 62-82, page 5), as well as the direct references to the relevant papers (Di Mascolo et al. 2019a, 2019b). References to any specific libraries and packages employed for actual implementation of the analysis methods is provided in the “Code availability” subsection at the end of “Methods”.

Extraction of the SZ signal with nested posterior sampling

The use of different SZ models to explain the difference between the measured data points and the model including the radio source alone (fig 1b) builds confidence in the detection. However, there are some issues the authors should address:

1. The errors on the radio source model shown in figure 1a (defined as “the standard deviation of the scatter within each uv bin due to combination of the elongated morphology of the radio galaxy and the asymmetry in the visibility patterns of the data” which is not clear to me), are never mentioned again in the analysis and do not seem to be propagated – if they are, the authors should make it clearer in the text. Indeed, since the errors are about 0.1 mJy large, taking them into account can lower the significance of the SZ detection.

We realised the graphical rendition of Fig. 1 could cause some graphical elements to be misinterpreted. In particular, the shaded region in the left panel does not represent an uncertainty on the radio source model or a statistical error, but reflects the inherent variation of the Fourier amplitude of such a model when considering different radial directions in the uv (i.e., Fourier) plane. This is the result of the asymmetric morphology of the radio galaxy (mostly elongated in the east-west direction), which in turn introduce a corresponding asymmetry in its Fourier equivalent (i.e., with the Fourier profile in the v direction being more extended than the one in the u direction). To better elucidate such an effect, we provide in Fig. R2.1 a comparison of the binned profiles computed over different quadrants of the uv plane. As

mentioned in the caption, such an effect combines with the asymmetric pattern of the visibility distribution, which enhances the spread in the Fourier amplitudes measured in each circular uv radial bin. The main goal is thus to show that the radio source model, although plotted for simplicity as a 1D profile, actually exhibits a broad azimuthal asymmetry as expected from the observed source morphology. In order to clarify this, we edited the caption, the figure legend (“scatter” to “asymmetry”) and adopted a hatched, step-like style (and, thus, different from the model uncertainties in panel 1b) for the azimuthal variation of the Fourier amplitudes.

2. Different numbers of compact components are used to model the radio source and, from my understanding, the “best” of these models is chosen before including the SZ contribution in the model (lines 82-92, page 5). If so, could the authors please make sure the results you cite about the SZ signal are marginalised over the number of compact components, or at least indicate how much the modelled SZ signal changes when source components are added or removed?

We ran a test by re-computing our “reference” SZ model (A10 UP) considering different number n of point-like components for the radio source model. We summarised our finding in a new subsection (“Dependence of the SZ significance on the number of point-like components”) in “Methods”, as well as in a new figure (Extended Data Fig. 3 in the current version of the draft). In summary, the addition of an SZ component is always statistically favoured in comparison to the radio source-only model, although the model parameters are observed to stabilise only for $n > 5$. Beyond $n = 8$ — our reference model — the inclusion of an additional point-like component is instead observed to degrade the statistical significance of the model. In addition to what already reported in the new subsection in “Methods”, we show in a new figure in Methods (Extended Data Fig. 4) a summary of the goodness of the various point-like components visualised in terms of the image-space residuals resulting from the subtraction of the respective radio source model. As noted in the main text, the model-subtracted data for $n \leq 5$ exhibit large residual signal, thus undermining the validity of the model in spite of any statistical estimators.

Could they also confirm that the “model uncertainties” cited in the caption of figure 1a correspond to the variance between models including various numbers of compact components?

As for the SZ model, we used the Bayesian Model Averaging to marginalize over the models with different number of compact components and re-evaluated the overall model variance (i.e., the 16th and 84th quantiles on the binned model) accordingly. We adapted the text in the caption of Fig. 1a to reflect this change. We note, though, that the uncertainties on the average radio model, also in this case, results to be too small to be visible in Fig. 1a. As shown in Fig. R2.2, the overall effect is a slight increase in the uncertainties on the model profiles compared to the reference $n = 8$ case.

3. Overall, a more complete description of the source model fit would be appreciated.

This is indeed an essential part of the discussion that we overlooked in the first place. Below is a list of edits introduced in the Method section “Nested posterior sampling” to solve this issue.

- We added a description of the specific choices adopted for the prior distribution of each model parameter. Such an information was also missing for the model component used to describe the SZ distribution, and was included at the end of the corresponding paragraph.
- In the subsection “Results” inside “Nested posterior sampling”, we provide a brief presentation of the results from the radio-source modelling. Briefly, the favoured model comprises eight components, six of which are distributed along the jet structure and have a negative spectrum in rough agreement with previous VLA analyses. The remaining two sources correspond to previously known protocluster galaxies (ERO 284 and HAE 229) and have positive spectral indices, consistent with a spectrum dominated by thermal dust emission.
- In a new table (Extended Data Tab. 1), we provide a summary of the best fit parameters for the selected set of point-like components.

Figure R2.2: Comparison between the models BMA-marginalized over the different number of point-like components and the reference, best-fit cases (i.e., for $n = 8$ point-like sources) when assuming an A10 UP and L15 8.0 pressure distribution. The median profile and credible intervals for the BMA models are denoted as solid lines and shaded regions, as dashed and dotted lines for the best-fit profiles.

- We included in Extended Data Fig. 1 (previously Fig. 4) markers (and labels) to denote the position of each point-like components, in order to provide a simple, visual description of their spatial distribution and to facilitate the comparison with the morphology observed for the radio jet structure.

In Fig. R2.3 we report the additional test we performed on the reference fluxes of the point-like sources to check whether the addition of an SZ component could result in any biases in their inference. Overall, we see no statistically significant variation between the case without any SZ component and the ones considering the different models for the ICM pressure distribution. We see a slightly larger effect (still within the statistical uncertainties) only on ID7 and ID8. These are the two components that fall closest to the SZ feature, and are thus inherently more affected than the rest of the radio source by any cross-contamination with the underlying SZ effect. In turn, the inclusion of an SZ term in the total model would naturally cause the flux amplitude to be larger than the model without any SZ component, as indeed observed in Fig. R2.3.

4. The effective significance cited in the main text (line 4, page 2), does not correspond to any of the values listed in Table 1 for the different SZ models considered.

As correctly noted but the referee, the value reported in the main text did not correspond to any of those reported in Tab. 1 (now Extended Data Tab. 2), as a result of mistyping. We corrected this. All the values provided for the effective significance should now be consistent throughout the manuscript.

5. The authors do not specify why the “best” and “worst” SZ models cited in the caption of figure 1b are considered as such.

As mentioned above, we reworded the introductory sentence in the caption of Fig. 1b in order to clarify this point. The “best” and “worst” SZ models correspond to the ones with the lowest and highest significance/Bayesian evidence (as summarized in Extended Data Tab. 2, previously Tab. 1).

Could the authors explicit how “broadly consistent with unity” the calibration scaling parameters are (within the 5% imposed? – line 35, page 6).

Figure R2.3: Comparison of the amplitude of the point-like components obtained in the radio-only modelling run and when considering different pressure models for describing the SZ effect. All the amplitude parameters remain practically constant across the different cases. The largest variation is observed for ID7 and ID8, the westernmost sources in the sample and the ones closest to the SZ feature.

We added in “Methods” all the best-fit values for scaling parameters. For consistency with the new organisation of the section, we moved this discussion at the end of the “Results” subsection.

As correctly hinted by the referee, they are in fact consistent with unity within the 5% uncertainty on the ALMA flux calibration. In fact, the final 16th-84th quantile ranges for all parameters are narrower than the input 5% standard deviation, suggestive of an optimal cross-calibration of the different data sets. For reference, we include in this report a corner plot showing the correlation of the scaling parameters with the mass term M_{500} and the point source amplitudes $p_{\{1,\dots,n\}}$, the model parameters that are most affected by any systematics in the flux calibration.

Theoretical interpretation and comparison to previous mass studies

Despite representing a large fraction of the “Methods” section, the discussion of the SZ-measured halo mass compared to previous measurements is barely mentioned in the main text. Although I understand no clear conclusion can be drawn, this analysis could probably be refined, and/or tentative explanations could be made clearer in the text (namely in the abstract).

As mentioned above, we included more details regarding the mass estimates both in the main text and in the abstract. Given the large uncertainties and the number of caveats required to correctly interpret the reported mass estimates, we still prefer to limit the bulk of the discussion to Methods.

1. Could the authors add a column to table 1 presenting the Y_{SZ} values for each model to support the discussion of page 7?

For general reference, we included in Extended Data Tab. 2 (previously Tab. 1) the values for the integrated Y_{SZ} within both r_{500} and $5r_{500}$.

2. In fig 2, the models with larger halo mass lead to stronger negative SZ signal (and supposedly a Y_{SZ} value closer to the one obtained from velocity measurements?). In figure 1b, you present SZ models that underfit the data points and lead to a lower mass than expected from other observations. Could the authors comment on this?

We would like to thank the referee for this comment as well as the suggestion to include the Y_{SZ} values in Extended Data Tab. 2 (previously Tab. 1), as it helped us identifying a subtle bug in the plotting script generating Fig. 1b (a missing c_{500} factor in the conversion from the physical radius r to the dimensionless

Figure R2.4: Marginalized posterior distribution for the mass parameter M_{500} (red column), the amplitudes $p_{\{1,\dots,n\}}$ of the point-like components (with $n = 8$ as for our reference radio model; black columns), and scaling parameters (blue columns) obtained under the assumption of a L15 8.0 model.

scale $r/r_s = c_{500} r/r_{500}$ when computing pressure models). Now the profiles show significant differences at large scales/short baselines, as one would expect for models with different masses (and, as correctly mentioned by the referee, different Y_{SZ} values). Also unexpectedly, they exhibit a remarkable agreement on the scales probed by the ALMA+ACA data. In Fig. R2.5 we provide an edited version of Fig. 1b showing the profiles for all the models employed in our study. For the sake of clarity while allowing for an easier comparison with Fig. 1b, we are plotting the confidence intervals only for the models A10 UP and L15 8.0 presented in Fig. 1b in the main draft. Nevertheless, we note that the uncertainties for all the models are comparable in amplitude to the ones for A10 UP and L15 8.0.

Figure R2.5: Comparison of the uv profiles for different models for the ICM pressure distribution. Except for the L15 REF profile (which, in fact, has the lowest Bayesian evidence in the sample), all the models exhibit a remarkable agreement on the ALMA scales, while deviate significantly one from the other at short baselines/large scales. Taken together, this suggest a dominant role of the ALMA data in constraining the SZ model.

This further solves (or, at least, drastically mitigate) the systematic underfitting issue highlighted in the previous version of this figure. The divergence of the SZ profiles at short uv distances and the tight agreement on the ALMA scales suggest that the residual shift is an effect of the limit of the selected analytic, spherically symmetric profiles in providing a correct description of the observed SZ signal. Further, this clearly shows that most of the constraining power is provided by the ALMA set, with a marginal contribution from the ACA measurements. To quickly test this, we repeated the modelling of one of the reference profiles (in particular, we arbitrarily consider L15 8.0, the model with the largest Bayesian evidence) using only the ALMA data. As shown in Fig. R2.6, the ALMA-only profile remains consistent with the ALMA+ACA model, even over the range of uv distances (i.e., angular scales) not probed by any of the ALMA data.

We finally note that repeating the same experiment considering only the ACA measurements would not allow for a meaningful one-to-one comparison with the full ALMA+ACA run. In fact, it is not possible to perform an ACA-only SZ reconstruction jointly with the full model for the radio source (i.e., as done for the reference runs reported in the text and the test discussed above on the ALMA data). This is a limitation caused by the reduced resolution of the ACA observations in comparison to ALMA, which contributes the most in driving the inference of the model parameters for the radio source. On the contrary, a single component would suffice to describe the ACA measurement of the extended radio signal. Nevertheless, for completeness, we show in Fig. R2.6 the profile obtained forcing the modelling set-up to be consistent with the one used for the main results provided in the text and the ALMA-only test (i.e., a total of eight RA-ordered point-like components with uniform priors on all their parameters, and an SZ component). This practically matches from the optimal model for the ACA data (i.e., the one comprising only one point-like component), but exhibits $\lesssim 10\%$ larger uncertainties as a consequence of the point-like components remaining practically unconstrained. Overall, the ACA profile is found to fall closer to the binned ACA points, seemingly reducing the minor discrepancy between the short-baseline ACA data and the profiles. Given the model assumption, we however caution to consider this result only as a general indication of the overall effect caused on the model reconstruction by the inclusion of the ACA measurements — i.e., an overall increase of the model flux on short baselines.

We want to stress that the mismatch between the mass inferred from dynamical considerations and the estimate from the SZ measurements is however highlighted not only by our data modelling. In fact, the

Figure R2.6: Comparison of the uv profiles for the L15 8.0 model inferred using ALMA only (red), ACA only (yellow), or both the ALMA and ACA measurement sets (dark green; this curve corresponds to the one reported in Fig. 1b for the selected model). As reported in the text and to allow for a meaningful comparison, the ACA-only model is inferred using the same modelling set-up as for the ALMA-only and ALMA+ACA runs, despite being sub-optimal. All the models agree within the 1σ limit over as well as beyond all the scales probed by ALMA and ACA.

comparison with the prediction from hydrodynamical simulations (Fig. 2) provide a similar result in terms of the mass required for a halo to exhibit an overall SZ footprint compatible with the ALMA observations. The best agreement is in fact observed for masses of $\sim (3 - 4) \times 10^{13} M_{\odot}$, i.e., a factor $\gtrsim 2$ smaller than expected from velocity dispersion. We added a brief statement in the main text (page 3, lines 35-38) to emphasize this.

3. The way the authors investigate the potential for several sub-haloes to contribute to the observed signal rather than only one halo is extremely interesting. Would it be possible to include a slightly more physical model of sub-halo mass distribution rather than subcomponents of equal mass (line 34, page 7) – maybe something as simple as a power law?

We tried considering a different mass distribution, but did not find any positive and constructive results. In particular, we constructed a basic model composed of n haloes with:

- masses distributed according to the sub-halo mass distribution provided in Giocoli et al. 2008;
- total mass equal to the one based on the velocity dispersion measurement reported in Shimakawa et al. 2014, consistently with the discussion in the main text;
- either the most or the least massive of the components constrained to have an integrated Y_{SZ} consistent with the one estimated from the ALMA+ACA data.

To avoid label switching and numerical instabilities, we further imposed an ordering condition on all the mass parameters ($M_1 < \dots < M_n$). As for the number of components n , we considered two cases:

- we fixed n to a given value and run the model reconstruction to check the effect of such a choice on the output masses;
- we set n to an arbitrary large value (in particular, we considered the cases $n = \{3, 10\}$) and imposed a sparsity prior on the masses of each component, in order to minimize the actual number of sub-haloes contributing to the total mass of the structure.

Independently on the assumptions on n and on the specific subhalo constrained to have a Y_{sz} consistent with the observed value, this modelling problem remains unconstrained. Below is a summary of the main issues we encountered.

- When forcing the model and one subhalo to have a total mass and integrated Y_{SZ} values *exactly* equal

to the dynamical mass and to the ALMA Y_{SZ} estimates, respectively, the constraint $n = 2$ results (not unexpectedly) in two model components of masses $M_1 = M_{500}$ and $M_2 = 1 - M_{500}$, with M_{500} equal to the one inferred from the Bayesian modelling (i.e., $M_1 \simeq 3.50 \cdot 10^{13} M_\odot$ when considering an A10 UP profile). For $n > 3$, the mass parameters for the haloes that are not bound to the Y_{SZ} measurements instead become heavily degenerate (see Fig. R2.7; such an outcome is independent of the specific subhalo used as reference for matching the observed Y_{SZ}).

- softening the constraints on the total mass and integrated Y_{SZ} (i.e., by considering their respective uncertainties by means of split-Gaussian likelihood terms) makes the ordering prior dominate the sampling of the posterior probability distribution. As a combination of the such a prior with the prescription on the sub-halo distribution cause all the mass parameters to assume similar values (although different enough to satisfy the ordering rule). However, turning the parameter ordering off causes the sampler to encounter severe numerical instabilities caused by label switching (see the corner plot in Fig. R2.8).
- The specific choice of the component to bind to the Y_{SZ} estimate is entirely arbitrary, but has the main issues of having major impact on the model inference. In particular, choosing the reference subhalo not to be the most massive in the sample, in combination with the sparsity prior introduce above, helps in breaking the severe degeneracy between the various mass estimates. However, this causes all the components constrained to have a mass smaller than the reference subhalo to be characterized by mass parameters either equal or order of magnitude smaller than the Y_{SZ} -constrained value. All the other components automatically tune to satisfy the constraint on the total mass, generally resulting in haloes with masses too large ($M_{500} \gtrsim 7 \times 10^{13} M_\odot$) not to be observed by ALMA+ACA.

All in all, applying a physically motivated subhalo mass function, although meaningful, would make the

Figure R2.7: Example of posterior probability distribution for subhalo mass search with Dirac- δ priors on the total mass and Y_{SZ} . The tilde in the parameter labels denotes the specific subhalo for which we impose the Y_{SZ} constraint. The subhalo mass parameters with no Y_{SZ} priors exhibit a strong degeneracy.

Figure R2.8: Example of posterior probability distribution for subhalo mass search with Dirac- δ priors on the total mass and Y_{SZ} without imposing an ordering prior on the mass parameters. The 2D marginalized posteriors clearly highlight the unstable behaviour of the sampler.

toy experiment of constraining the number of subhaloes highly unstable and hard to interpret. We thus decided not to provide a detailed report on the above results in the manuscript, but to still briefly mention this in the draft.

4. Would it be possible to allow for more than one SZ (pressure) component in the Bayesian analysis? The strength of the argument made on line 53 of page 7 about the posterior velocity distribution not being bimodal is dubious to me if several components are not allowed in the analysis in the first place.

It is indeed possible to allow for multiple SZ components and we tested for such a scenario. The related results are discussed in the comments to the Referee Report #1 (see, in particular, Fig. R1.3). We further provide in a new subsection (“Multiple SZ components”, inside “Nested posterior sampling” in the “Methods” section) a general presentation of our findings when allowing for more SZ components, as well as for ellipticity.

Typos & general comments

1. Please give the definition of the log-evidence used in this work.

We added a brief definition of Bayesian evidence at page 2 (line 30-32). As mentioned above, we provide also a description of all the other statistical estimators used in our work.

2. Could the authors explain in the text their conclusion about the virialization of the gas? It does not seem to me that the statement “the identification of virialized subhaloes through a spectroscopic characterization is ineffective” made by the authors (line 25, page 3) is properly backed.

The referee is correct. In fact, the reported statement is not providing the correct message, i.e., that the previous spectroscopic studies of the Spiderweb complex did not allow to identify sub-haloes. As such, we first substituted the word “virialized” with “distinct”. More in general, we removed all mentions to the potential virialization state of the system. First, the quality of the available data is not providing for inferring any strong constraints on the dynamical state of the structure or any of its parts (we adapted our conclusions to better emphasize this). Second, the limited information we infer from the multi-wavelength data on the physical state of the Spiderweb protocluster highlights that the multi-phase gas withing is experiencing a dynamically active, merging phase. As such, an actual virialization of the structure is automatically excluded.

3. A flat prior can have strong effects on your posterior distributions and be far from “uninformative” (line 91, page 5).

This is indeed correct and the result of the common use of the misnomer “uninformative” in relation to uniform priors. Not only, the non-uninformative nature of a uniform distribution is particularly relevant when the uniform probability assumption is combined with the ordering prescription. We updated this definition accordingly.

4. Typos: lines 7 and 8, page 3; line 81 page 11

We thank the referee for noting these. We corrected the typos.

5. The abstract is clear, well-written, and showcases the importance of this detection well. However, the last sentence, related to the theoretical interpretation, is vague, unconvincing, and difficult to relate to the analysis presented in the main text.

We agree with the referee and, as mentioned above, we modified the conclusive statements of the abstract to better reflect the content of the paper.

6. It does not seem that the availability of the ACA data used in this work is listed.

The availability statement for the ACA data is provided at the end of the “Methods” part. The Band 3 and Band 4 ACA measurements are part of the 2016.2.00048.S and 2018.1.01526.S ALMA projects, respectively.

7. Could the authors explain how the protocluster considered in this work is archetypal?

We decided to remove any of such references. The “archetypal” aspect of the Spiderweb galaxy is mostly an historical consequence, coming from the extensive (and probably unique) observational coverage collected since the discovery of the system. This has made the system the most studied protocluster complex, and, such, a key reference for protocluster studies in general. In fact, the Spiderweb protocluster shows many peculiarities (first of all, the extended hybrid morphology radio jets), which effectively makes the system more of a spectacular, rather than exemplary case of a protocluster system.

Reviewer Reports on the First Revision:

Referee #1 (Remarks to the Author):

I thank the authors for their detailed and clear responses. I am satisfied that the current manuscript is acceptable for publication in Nature.

Referee #2 (Remarks to the Author):

Dear editor, dear authors,

First, I would like to thank the authors for giving thorough answers to my comments and for extensively modifying the manuscript accordingly. I particularly appreciate the explanations given on the signal shift, with Figs. R2.5 and R2.6 being enlightening.

The additional details given on the source shift and the investigation of the impact of including a different number of SZ or point-source components will be particularly useful to readers and make the detection all the more convincing. The definitions of the statistical terms used that the authors added to the text are also helpful.

I can now recommend the manuscript for publication.

I have a few additional questions, some out of scientific curiosity and some asking for precisions. However, I do not require the authors to answer them at this point. You can find these comments below.

Kind regards.

Extraction of the signal with nested posterior sampling

Comment #2: On the number of source components

I appreciate the authors adding a new subsection to investigate the impact of the number of source components on their results. It is reassuring to see the impact is low and that the choice of $n=8$ has strong grounds, both from a physical (ED Fig. 4) and a Bayesian (ED Fig. 3) point of view.

About ED Fig. 3, on the first two panels, I seem to understand from the caption and the axis labels that there should be two lines, e.g., for panel 1, one line for σ^{pts} and one for $\sigma^{\{\text{pts}+\text{sz}\}}$ but I can only see one? Additionally, how can the variance of the effective significance (panel 2) be negative ($n=9$) if it is defined as a square root?

Comment #3: On the description of the source model

I appreciate the authors adding markers to denote the different point-like components of the fit, in ED Fig. 1.

Analysing Fig R2.3, the authors claim that changing the SZ model has a stronger effect on the final two source components (ID7 & ID8) than others, and if their physical argument makes sense ('these are the two components falling closest to the SZ feature'), it is far from obvious to me from the figure. For all components, it seems the central values do not shift by more than $\sigma/10$ between SZ models. Could the authors explain their reasoning?

Theoretical interpretation and comparison to previous mass studies

Comment #2: On underfitting

I would like to thank the authors for investigating the shift in detail and considering various options for the mass distribution between sub-haloes. I have one question: What are the parameters impacted when removing the ACA data points from the fit? They must be the ones governing the large scales of the SZ profile and so will be most impacted by new data.

Additionally, in lines 6-7, page 2 of the manuscript, the authors state 'ACA data [are] most sensitive to the SZ signal thanks to its ability to recover large angular scales' which is contradictory with the argument made with Fig. R2.6 that the ALMA data is most constraining for the SZ model.

Author Rebuttals to First Revision:

In the report below, the comments from Referee 2 are reported in red, our responses in black.

First of all, we would like to thank the referee for all comments and feedback, which provided important insights for improving our manuscript.

Extraction of the signal with nested posterior sampling

Comment #2: On the number of source components

I appreciate the authors adding a new subsection to investigate the impact of the number of source components on their results. It is reassuring to see the impact is low and that the choice of $n=8$ has strong grounds, both from a physical (ED Fig. 4) and a Bayesian (ED Fig. 3) point of view. About ED Fig. 3, on the first two panels, I seem to understand from the caption and the axis labels that there should be two lines, e.g., for panel 1, one line for σ^{pts} and one for $\sigma^{\{\text{pts}+\text{sz}\}}$ but I can only see one?

There are indeed two lines in the first two panels of ED Fig. 3. However, the dynamic range of the plots make the respective σ^{pts} and $\sigma^{\{\text{pts}+\text{sz}\}}$ lines too close to be distinguished. We thought about adding separate panels for each line, but, in our opinion, this would not have provided any insightful information. Anyway, to better highlight this issue, we adapted the caption of ED Fig. 3.

Additionally, how can the variance of the effective significance (panel 2) be negative ($n=9$) if it is defined as a square root?

We thank the referee for pointing this issue out, as it highlights negligence in our notation. Given a log-evidence difference $\Delta \log Z$, the correct expression for computing σ_{eff} is in fact $\sigma_{eff} = \text{sgn}(\Delta \log Z) \cdot \sqrt{2|\Delta \log Z|}$, where the sgn function is used to describe any potential reduction of the Bayesian evidence in terms of a negative variation of the effective significance.

We modified the definitions throughout our manuscript accordingly.

Comment #3: On the description of the source model

I appreciate the authors adding markers to denote the different point-like components of the fit, in ED Fig. 1.

Analysing Fig R2.3, the authors claim that changing the SZ model has a stronger effect on the final two source components (ID7 & ID8) than others, and if their physical argument makes sense ('these are the two components falling closest to the SZ feature'), it is far from obvious to me from the figure. For all components, it seems the central values do not shift by more than $\sigma/10$ between SZ models. Could the authors explain their reasoning?

The referee is correct, and, in fact, considering different SZ models is not introducing any major variation in any of the point-like sources. The "stronger effect" we are mentioning when referring to ID7 and ID8 is however the one we observe when considering no SZ term at all (i.e., the leftmost data point labelled "no SZ") versus any of the reference SZ models (which, in fact and as correctly mentioned by the referee, agree with each other at the $\sigma/10$ level). Nevertheless, as mentioned in the previous report, the values for ID7 and ID8 for the no-SZ model are still in statistical agreement with the SZ cases.

Theoretical interpretation and comparison to previous mass studies

Comment #2: On underfitting

I would like to thank the authors for investigating the shift in detail and considering various options for the mass distribution between sub-haloes. I have one question: What are the parameters impacted when removing the ACA data points from the fit? They must be the ones governing the large scales of the SZ profile and so will be most impacted by new data.

Indeed, this is generally the case. Adding more ACA data results in better constraints on the more diffuse/large-scale component of the SZ distribution (and, of course, the other way round when excluding the ACA measurements). From a physical point of view, this would result in reduced uncertainties on the SZ profile at large radii, where the SZ profile is expected to be shallower than in the core region. In terms of model parameters, when considering any of the several flavours of the gNFW pressure model, improved large-scale data would improve the inference of the profiles slopes at intermediate and large radii (generally denoted as β and α). Nevertheless, the more flexible the parametrisation, the more the final result will be affected by the numerous degeneracies inherent to the gNFW description (e.g., between pressure normalisation, core radius, and profile slopes).

In the specific case of the analysis of the Spiderweb data, we however considered self-similar gNFW formulations, with fixed radial slopes and with the pressure normalisation constrained to be function of the core radius. This allows for breaking (or at least limiting) the dominant degeneracies in the gNFW pressure models, improving the model convergence and guaranteeing a more straightforward interpretation of the modelling outcome.

The net result of removing the ACA data is thus an overall (but mild) increase of the uncertainties for all the model parameters. The strongest effect is observed for the mass parameters M_{500} , with an average increase of $\sim 20\%$ in its uncertainties (as measured from the 16th and 84th quantiles of the posterior distribution for M_{500}). In the specific case of M_{500} , the additional effect is a lower (but still statistically significant) estimate for the best-fit value of M_{500} compared to the ACA+ALMA case (e.g., in the case of the L15_8.0 profile, our “best” model, the ALMA-only estimate is $4.84^{+0.60}_{-0.48} \cdot 10^{13} M_{\odot}$, compared to the reported ACA+ALMA value of $5.46^{+0.47}_{-0.55} \cdot 10^{13} M_{\odot}$).

Additionally, in lines 6-7, page 2 of the manuscript, the authors state ‘ACA data [are] most sensitive to the SZ signal thanks to its ability to recover large angular scales’ which is contradictory with the argument made with Fig. R2.6 that the ALMA data is most constraining for the SZ model.

We agree with the referee that the current wording might sound contradictory. The message we would like to convey is that ACA reaches shorter baselines and, in turn, larger angular scales than ALMA, better covering the spatial range relevant for the SZ effect. The limited sensitivity of the ACA data in comparison to the ALMA measurements is instead making the latter driving the overall modelling process. We adapted the corresponding statement to “probing the SZ signal better than ALMA thanks to its ability to recover larger angular scales”, removing any confusing mention to sensitivity.